

# Topological data analysis for revealing dynamic brain reconfiguration in MEG data

Ali Nabi Duman[1] and Ahmet E. Tatar[2]

[1] Department of Mathematics, King Fahd University of Petroleum and Minerals, Dhahran, Saudi Arabia
[2] Center for Information Technology, University of Groningen, Groningen, Netherlands

## ABSTRACT

In recent years, the focus of the functional connectivity community has shifted from stationary approaches to the ones that include temporal dynamics. Especially, non-invasive electrophysiological data (magnetoencephalography/electroencephalography (MEG/EEG)) with high temporal resolution and good spatial coverage have made it possible to measure the fast alterations in the neural activity in the brain during ongoing cognition. In this article, we analyze dynamic brain reconfiguration using MEG images collected from subjects during the rest and the cognitive tasks. Our proposed topological data analysis method, called Mapper, produces biomarkers that differentiate cognitive tasks without prior spatial and temporal collapse of the data. The suggested method provides an interactive visualization of the rapid fluctuations in electrophysiological data during motor and cognitive tasks; hence, it has the potential to extract clinically relevant information at an individual level without temporal and spatial collapse.

# INTRODUCTION

The functional connectivity studies focusing on the co-activation of spatially separated brain regions have been shown to be fruitful in identifying special features of neural connectivity during resting state and cognitive tasks (*Bastos & Schoffelen, 2016*; *O'Neill et al., 2015*; *Friston, 2011*). These results open the way for using functional connectivity as a biomarker for clinical diagnosis which, for example, measures illness severity (*Brookes et al., 2016*) and predicts the response to clinical intervention (*Carbo et al., 2017*). Despite their success, most of these methods neglect the temporal variations during neural processes (*Sporns, 2013*; *Hutchison et al., 2013*). Time-resolved approaches, on the other hand, can provide a crucial understanding of how the information is processed in the brain as functional connectivity may alter within several hundred milliseconds of a cognitive experiment (*Smith et al., 2012*; *Chang & Glover, 2010*; *Tagliazucchi et al., 2012*; *Antonakakis et al., 2020*).

The most common neuroimaging modality to study dynamic functional connectivity has so far been functional Magnetic Resonance Imaging (fMRI). However, it is a challenge to monitor the rapid changes in network dynamics within a short time interval using fMRI, as it only detects a proxy of neuronal activity (*i.e.,* hemodynamic signals) in the brain. In

Corresponding author
Ali Nabi Duman,
aliduman@kfupm.edu.sa

this case, the most reasonable time required to calculate connectivity is 30 s (*Allen et al., 2012*). On the other hand, electrophysiological modalities (*i.e.,* ECoG, EEG, MEG) address this issue with their high temporal resolution and decent spatial scales. Recently introduced MEG/EEG analysis methods (*de Pasquale et al., 2010*; *Liu et al., 2010*) enable us to explore the variations in connectivity dynamics which eventually leads us to the determination of biomarkers in clinical applications.

The sliding window method is a simple and common approach for measuring dynamic connectivity in electrophysiological modalities as it is compatible with most of the static connectivity measures (*Brovelli et al., 2017*; *O'Neill et al., 2015*; *O'Neill et al., 2017*; *Carbo et al., 2017*; *Lee et al., 2017*; *de Pasquale et al., 2010*; *Doron, Bassett & Gazzaniga, 2012*; *Bassett et al., 2011*; *Antonakakis et al., 2020*). In this approach, the connectivity is assessed over the time windows of a fixed width which are shifted by a fixed step from the beginning to the end of the experiment. The main limitation with this approach is choosing the window length: While the results with a short window width are affected by noise, the ones with a long window width ignore the rapid fluctuations in the connectivity. Moreover, a fixed window width is not appropriate for the experiments with varying timescales of fluctuation. Hidden Markov Models (HMM) address this issue by measuring the connectivity of the data over aggregated intervals corresponding to certain states, which are characterized by the properties of the source level signal (*Baker et al., 2014*; *Vidaurre et al., 2016*). A limitation of this method is scalability in the case of a high number of time points and subjects. It is also more suitable to apply to a small number of regions at a time due to possible overfitting resulting from a large number of regions (*Vidaurre et al., 2018*). Examination of microstates in resting-state or during task is another well-established method of analyzing temporal brain activity. Microstates are a series of time intervals during which the scalp potential field's configuration is quasi-stable (*Michel & Koenig, 2018*; *Khanna et al., 2015*; *Milz et al., 2016*). Similar to other discrete methods, microstate analysis collapses the signals on the time scale losing the possibly valuable information related to the high temporal resolution of MEG data. Different brain patterns might result in the same microstates as the inverse of the microstates are not unique.The effectiveness of these discrete approaches to quantify the brain dynamics is still under discussion among the functional connectivity community (*Preti, Bolton & Ville, 2017*). An alternative to avoid the discretization on the temporal scale is the high temporal measures of connectivity exploiting wavelet transform (*Dhamala, Rangarajan & Ding, 2008*) or Hilbert transform (*Breakspear, 2002*). Although the high temporal measures are applicable to individual data and capture the connectivity instantaneously, they require averaging, for example, over the trials of the task to increase robustness. Collapsing or averaging the data in time or space makes it difficult to derive the complete picture of how the brain connectivity continuously evolves during rest and task (*Saggar et al., 2018*). To improve translational outcomes, it is important to explore new computational methods that avoid averaging data across individuals, time, or space.

The use of artificial neural networks (ANN) in scientific studies has been increased in recent years (*Chiniforooshan Esfahani, 2023*). This trend can also be noticed in brain dynamics research. Deep neural network models are used to improve electrophysiological

source imaging of spatiotemporal brain dynamics (*Sun et al., 2022*). Physics-informed neural networks are applied to investigate molecular transport in the human brain using MRI images (*Zapf et al., 2022*) and to provide high-resolution maps of velocity, area, and pressure in the entire brain vasculature from Transcranial Doppler ultrasound data (*Sarabian, Babaee & Laksari, 2022*). Various graph neural network architectures designed to forecast brain activity based on models of spatiotemporal brain dynamics are compared in *Wein et al. (2022)*. Recurrent neural networks are used to predict feature-evoked response sequences from fMRI data (*Güçlü & van Gerven, 2017*). The new MEG datasets are also emerging to train and test brain-based ANN models. For example, a narrative comprehension MEG data representing rich variety of temporal dynamics is provided to test ANN based current natural language processing models against brain data (*Armeni et al., 2022*). ANNs have recently been utilized to improve disease diagnosis and classification. Deep neural networks using resting state EEG data of elderly individuals is proposed as a diagnosis tool for preclinical Alzheimer's disease (*Park et al., 2022*). Attention deficit and hyperactivity disorder (ADHD) classification with EEG and ANNs is studied in *Martínez González et al. (2022)*. Another recent application is the detection of Parkinson's disease through resting-state EEG based deep neural networks (*Shaban & Amara, 2022*). While ANNs are black-box tools with high accuracy for classification and prediction, their results are hard to interpret due to the complex underlying algorithms. More interactive and inherently interpretable tools are necessary for clinical use.

To address the above issues, we utilize a topological data analysis (TDA) technique called Mapper (*Singh, Memoli & Carlsson, 2007*; *Carlsson, 2014*) using MEG data from the Human Connectome Project (*Larson-Prior et al., 2013*). Due to its low sensitivity to noise and coordinate and deformation invariance features, the mathematical graph obtained from Mapper for each subject's MEG data can be visually and graph-theoretically explored making it accessible for clinical use.

Mapper has previously been applied to longitudinal MRI revealing two large subgroups within the population ($n = 52$) of children diagnosed with Fragile X syndrome. Mapper is shown to be promising in brain dynamics analysis for fMRI on an individual level ($n = 1$) making it valuable for translational studies. The mesoscale graph invariants (*i.e.,* modularity and core–periphery) of the output mathematical graph representation not only predict task performance but also differentiates the time points during evoked tasks and resting state by locating them at the core and periphery of the Mapper graph, respectively (*Saggar et al., 2018*). In a succeeding work, new Python-based interactive visualization tools are provided to examine Mapper graphs (*Geniesse et al., 2019*). Limitations of Mapper such as the need of dimensionality reduction and exploration of vast parameter space are addressed *via* NeuMapper framework in *Geniesse, Chowdhury & Saggar (2022)*. Mapper has detected a transition state of the brain between different neural configuration from resting-state fMRI data (*Saggar et al., 2022b*). In another work, Mapper has provided an evidence that a greater differential engagement of brain activity was achieved using methylphenidate during an n-back working memory fMRI task (*Saggar et al., 2022a*). In *Zhang, Chowdhury & Saggar (2023)*, dynamical systems features are extracted from Mapper graphs to bridge the gap between data-driven models and mechanistic dynamical systems models.

Here, we apply the Mapper algorithm to the temporal dimension of MEG data, which, unlike fMRI, provides direct measurement of whole-brain activity with richer temporal information. The mesoscale graph invariants of the Mapper graph are shown to be effective in differentiating data collected during working memory, story/math, and sensor–motor experimental paradigms. When these paradigms are compared pairwise, it is found that

- the centrality scores of the working memory task are significantly higher than the centrality scores of the story/math task;
- the centrality scores of the sensory-motor task are significantly higher than the centrality scores of the story/math task;
- there is no significant difference between the centrality scores of the sensory-motor task and the centrality scores of the working memory task;

pointing out the high stability in the temporal brain functional during the high demanding tasks. These results partially agree with the fMRI results (*Saggar et al., 2018*). Additionally, for the working memory and the story/math tasks where the performance is measured and timed, it is shown that there is weak negative and non-significant correlation between the community structure of the graph and the response time. A stronger but statistically weak correlation is also noticed in the fMRI study (*Saggar et al., 2018*). In summary, Mapper provides an interactive visualization of the rapid fluctuations in electrophysiological data during rest and cognitive tasks; hence, it has the potential to extract clinically relevant information at an individual level without temporal and spatial collapse.

## METHODS

### Ethics statement
This article utilised data collected for the HCP (*Van Essen et al., 2012*). The scanning protocol, participant recruitment procedures, and informed written consent forms, including consent to share deidentified data, were approved by the Washington University institutional review board (*Van Essen et al., 2012*). The IRB approval number is FEB-20220810-13614.

### Subjects and data
The data we use in this research is the human non-invasive resting state and task Magnetoencephalography (MEG) data set which is publicly available from the Human Connectome Project (HCP) consortium (*Larson-Prior et al., 2013*). It is acquired on a Magnes 3600 MEG (4D NeuroImaging, San Diego, CA, USA) with 248 magnetometers and 23 reference channels at the sampling rate of 2034.5101 Hz. The data is available for a total of 100 subjects each performing three experimental paradigms: Sensory-motor, Working memory, and Story/Math.

The tasks in the sensory-motor paradigm involve the execution of a simple hand or foot movement. Which limb on which side is instructed by a visual cue, which serves to pace the movement. The working memory paradigm is similar or identical to the corresponding task acquired during fMRI imaging of HCP. Here, the participants have to remember

the occurrence of $n$-back previously shown item (with $n = 0$ and $n = 2$) with the items being either tools or faces. Data are segmented to the onset of the non-target item (WM task). The story/math paradigm is the same as that is used in the fMRI component of the HCP (*Barch et al., 2013*). Participants listen either to auditory narratives of around 30 s duration or matched-duration simple arithmetic problems followed by a 2-alternative forced choice question (*Binder et al., 2011*). Subjects respond by right-hand button press (index or middle finger).

For every subject, all the paradigms consist of two experimental runs. A run consists of blocks of tasks, a block consists of several trials of the same task with a fixation period between the trials, and finally, a trial consists of a baseline and a stimulus. We shall also note that not all data is available for each subject.

For a single subject, the data acquired by Magnes 3600 MEG is processed as follows:

- Noisy channels with a high variance ratio and correlation to neighboring channels are detected and removed from further analysis.
- The bad channels and segments are removed with iterative independent component analysis (ICA) using spatial and temporal criteria (*Mantini et al., 2011*).
- Using ICA, independent components (ICs) are classified as 'brain' or 'noise' using six parameters: correlation between IC signals, the correlation between power time courses, the correlation between spectra, and three additional parameters derived from both spectral and temporal properties. Physiological artifacts are identified as magneto- and electro-cardiogram, eye movements, power supply bursting, and $1/f$-like environmental noise. The details of this step can be found in *Larson-Prior et al. (2013)* and *Mantini et al. (2011)*.

We note here two of its stages that are related to our preprocessing explained in the next section. First, the sampling rate is lowered to 506.6275 Hz, and second, the data from noisy channels are removed. As a result, the time points are reduced by 25% and the channels across the two runs might not be identical.

## Preprocessing

We process the data further per subject and per paradigm. We lower the sampling rate down to 256 Hz. We remove all the time points corresponding to the fixation trials and the baseline keeping only the stimulus ones as well as the time points with missing values. Moreover, to concatenate the data from the two runs, we also remove the non-common channels across the runs. As a result, a subject for a given paradigm is represented by a matrix of the form (# channels) $\times$ (# time points). The number of channels is changing between 200 to 248 depending on the subject, and the number of time points is ranging between 350,000 and 400,000.

At this point of the preprocessing, each subject is associated with three matrices, one for every experimental paradigm. As we want to compare the paradigms pairwise, in the next stage we concatenate these matrices two by two. This way, each subject is still associated with three matrices corresponding to the cases:

- Working memory and story/math;

- Working memory and sensory-motor;
- Story/math and sensory-motor.

These matrices are concatenated in the above order across the common channels in the paradigms. Even though the concatenation results in the loss of some channels, the loss is not significant and the resulting concatenated matrix has still 200 to 250 channels. In the meantime, the number of time points in the concatenated matrix is almost doubled and ranges between 700,000 and 800,000. In the end, the concatenated matrix is transposed so that the time points are on the vertical axis and the channels are on the horizontal axis.

Finally, we note that this vectorization of the MEG images causes the loss of the locations of channels relative to each other. However, it is shown in various studies that this process does not affect the success of the machine learning methods (*Bray, Chang & Hoeft, 2009*).

## Topological data analysis: Mapper

Most of the network neuroscience studies utilize simple graphs focusing on dyadic connection ignoring higher-order interactions that could be crucial to extract insight across multiple scales (*Torres et al., 2021*). The existence, quantification, and comparison of these higher-order interactions necessitate the use of more advanced mathematical structures that can be studied by topological data analysis(TDA), which uses techniques from algebraic topology and computer science to analyze data sets (*Centeno et al., 2022*). Analysis of these non-dyadic relations makes it possible to deal with the open problems in network neuroscience (*Andjelković, Tadić & Melnik, 2020*; *Billings et al., 2021*; *Guo et al., 2021*; *Helm, Blevins & Bassett, 2021*; *Patania et al., 2019*; *Santos et al., 2019*; *Saggar et al., 2018*).

In our study, we adopt a TDA method called Mapper, which is a successful structure discovery and visualization technique for the exploration of high-dimensional data. The resulting mathematical graph from Mapper is a highly compact representation of the complex data revealing insightful coordinate-free visualization. Introduced in 2007 (*Singh, Memoli & Carlsson, 2007*), its application to biological data sets includes but is not limited to disease association, RNA folding, viral evolution, and immunology (*Chan, Carlsson & Rabadan, 2013*; *Nielson et al., 2015*; *Li et al., 2015*). Hence, its application to brain dynamics is promising as shown in the earlier studies (*Geniesse, Chowdhury & Saggar, 2022*; *Geniesse et al., 2019*; *Saggar et al., 2018*; *Patania et al., 2019*).

The construction of a Mapper graph from a point cloud is illustrated in Fig. 1: (i) The first step is the choice of a *filter* which assigns one (or more) values to each data point in the point cloud. The filter values can be height, coordinate values, a measure of centrality, or output of any data mining algorithm such as PCA, SVD, SNE, *etc.* (ii) The next step is to cover all possible filter values with overlapping intervals (or regions depending on the dimension of the filter). Three color-coded intervals covering the range of the height function are shown in Fig. 1. (iii) Next, the points whose filter values fall in the same interval (or region) clustered using a clustering algorithm such as hierarchical clustering, k-nearest neighbor (KNN), or single linkage clustering. For clustering, one can use any metric including correlation, Euclidean, $L_1$ or $L_\infty$ metrics. (iv) The nodes representing

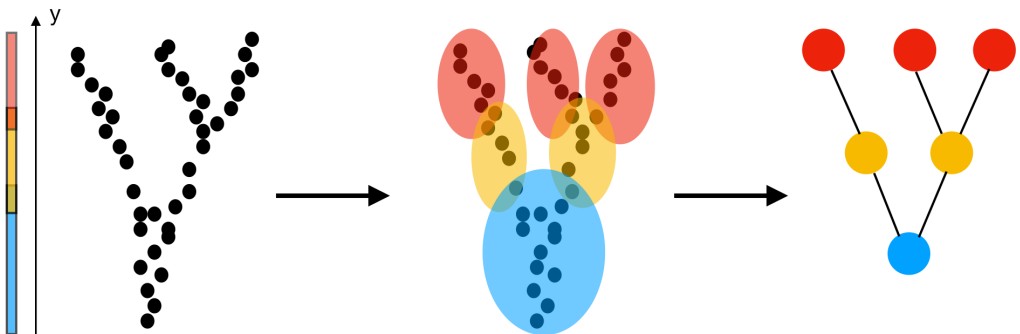

**Figure 1** **Construction of a Mapper graph from a point cloud.** The filter function is chosen as the $y$-coordinate (height function). After clustering the points whose height fall in the same interval, the vertices representing the clusters are joined by an edge if they have a common a point. The geometry of the large number of points is represented by few edges end vertices.

the clusters are finally connected by an edge if the underlying clusters have a non-empty intersection.

The parameters of the algorithm are the number of intervals (or regions), the overlapping percentage, the distance metric, and the clustering algorithm. A high number of intervals increases the number of nodes in the final visualization; hence, defeats the purpose of having a highly compact representation. While an increase in the overlapping percentage results in a high number of edges and increases complexity, a low overlapping percentage produces disconnected clusters and misses the information about variation in the data due to underlying continuous filter values.

We apply the Mapper algorithm to the MEG data from HCP. We use the open source KeplerMapper Python package (*Veen et al., 2019*) to generate Mapper graphs from the minimally processed data. Our goal is to trace the brain activation patterns of each participant during working memory, story/math, and motor tasks. As explained above, the data is concatenated pairwise before entering the Mapper algorithm. It is not uncommon to use concatenated data to estimate task-state functional connectivity and brain networks (*Richiardi et al., 2011*; *Hsu et al., 2014*; *Freeman, Donner & Heeger, 2011*; *Liu et al., 2014*; *Mokhtari & Hossein-Zadeh, 2013*; *Zhu et al., 2017*). Moreover, comparative studies show that functional connectivity for the concatenated data was both qualitatively and quantitatively similar to that of continuous data during rest (*Fair et al., 2007*; *Gavrilescu et al., 2008*; *Cheng et al., 2015*) and task (*Zhu et al., 2017*).

The input data to Mapper is the (# time points) $\times$ (# channels) dimensional matrix prepared by the preprocessing explained in Preprocessing.

We choose the Euclidean metric as a similarity measure between the vectors in the time-space (*i.e.,* column vectors) of the input matrix. The euclidean metric is suitable in this setting as the ranges and means of the data columns do not vary significantly (*Nielson et al., 2015*). The choice of Euclidean metric is also proven to be successful in the previous applications of neuroimage data: In *Romano et al. (2014)*, Mapper with Euclidean metric applied on sMRI data reveals high and low functioning neuro-phenotypes within Fragile X

Syndrome. In *Kyeong, Kim & Kim (2017)*, the method on fMRI data identified two unique subgroups of ADHD using the Euclidean metric. In another fMRI study, a data-driven search for different metrics indicates that the Euclidean metric best localizes outcome measures (*Madan et al., 2017*).

In the next step, the similarity information determined by the Euclidean metric is transformed into a low dimensional representation using a non-linear filter called t-SNE (*Hinton & Roweis, 2002*), which maintains the local geometry existing in the original time-space unlike more conventional linear filters such as PCA (*van der Maaten & Hinton, 2008*; *Saggar et al., 2018*). Multivariate and non-linear characteristics of inter-regional interactions suggest the use of non-linear methods such as t-SNE (*Reinen et al., 2018*; *DiCarlo, Zoccolan & Rust, 2012*). The data which is first reduced into two dimensions by t-SNE is then divided into overlapping bins. Following the earlier practice (*Lum et al., 2013*), the time points in each bin are further clustered using single linkage clustering with the Euclidean metric, which is computationally more efficient compared to the other clustering methods, and which does not require an initial number of clusters.

The common practice in Mapper applications is to test a large grid of parameters (*i.e.,* overlapping percentage and number of intervals) to find the most stable graphs. Even though the stability of the Mapper algorithm under various parameters was studied before under certain conditions (*Carrière & Oudot, 2018*; *Kalyanaraman, Kamruzzaman & Krishnamoorthy, 2017*), we analyze several parameters of the algorithm to ensure the reliability of our result.

## Centrality and community structure analysis

The mathematical graphs obtained from Mapper can be investigated by focusing its structures on different scales. It is established through many applications that the intermediate (mesoscale) structures identify certain characteristics that the local scale analysis of nodes (or edges) and global level of summary statistics are unsuccessful to detect. In this article, we concentrate on centrality and modularity mesoscale properties.

Centrality analysis of a network reveals the most important nodes (or edges) based on a quantification of node-node or node-edge relationships. The centrality of a node can be perceived as the communication ability with the other nodes or the closeness to the other nodes (*Estrada, 2011*). The centrality structure provides a perspective to comprehend how the brain states evolve during the ongoing task. The nodes with high centrality in the Mapper graph contain the most common brain activation patterns during the task. Here, we utilized four centrality measures: degree, eigenvalue, betweenness, and closeness. Based on the visual evidence from the box plots Figs. 2, 3, and 4 and the similar findings in *Saggar et al. (2018)*, we claim for any one of those four centrality scores that

$H_{WS}$: The mean of the centrality score of the nodes dominated by the working memory paradigm time points is greater than the mean of the centrality score of the nodes dominated by the story/math paradigm time points.

$H_{WM}$: The mean of the centrality score of the nodes dominated by the sensory-motor paradigm time points is greater than the mean of the centrality score of the nodes dominated by the working memory paradigm time points.

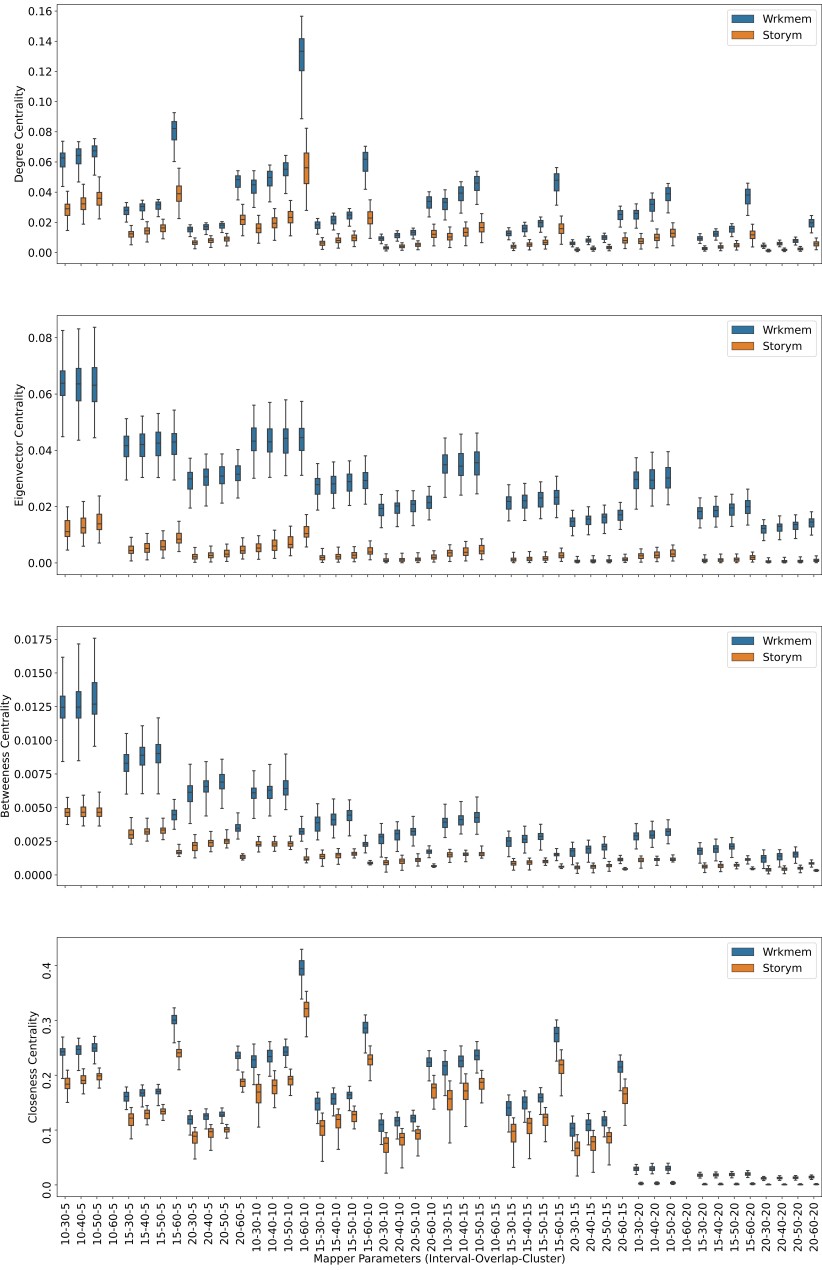

**Figure 2   Working memory *vs.* Story/math.** Box plots showing (from top to bottom) the degree, eigenvector, betweenness, and closeness centrality scores of the nodes of the Mapper graphs under different parameters. Parameters $(10-60-5), (10-60-15), (10-60-20)$ are missing due to high complexity of the calculations and the limitations of our workstation.

$H_{SM}$:   The mean of the centrality score of the nodes dominated by the sensory-motor paradigm time points is greater than the mean of the centrality score of the nodes dominated by the story/math paradigm time points.

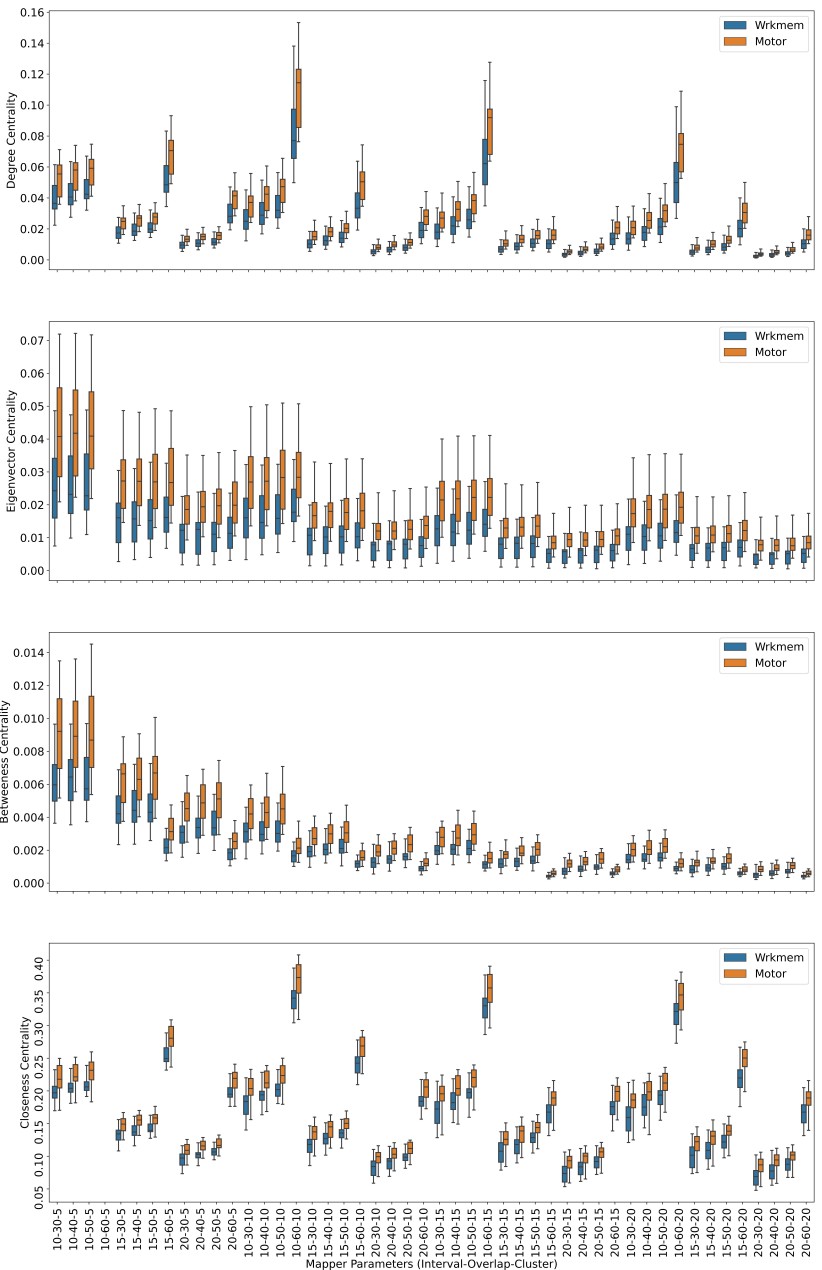

**Figure 3** **Working memory *vs.* Sensory-motor.** Box plots showing (from top to bottom) the degree, eigenvector, betweenness, and closeness centrality scores of the nodes of the Mapper graphs under different parameters. Parameter $(10 - 60 - 5)$ is missing due to high complexity of the calculations and the limitations of our workstation.

Modularity is one of the most commonly used metric to detect and characterize the community structure of the networks. Detecting communities in brain networks, are useful to identify the sub-networks that correspond to specialized functional components (*Sporns & Betzel, 2016*). In this article, we use modularity as defined in *Newman (2006)*.
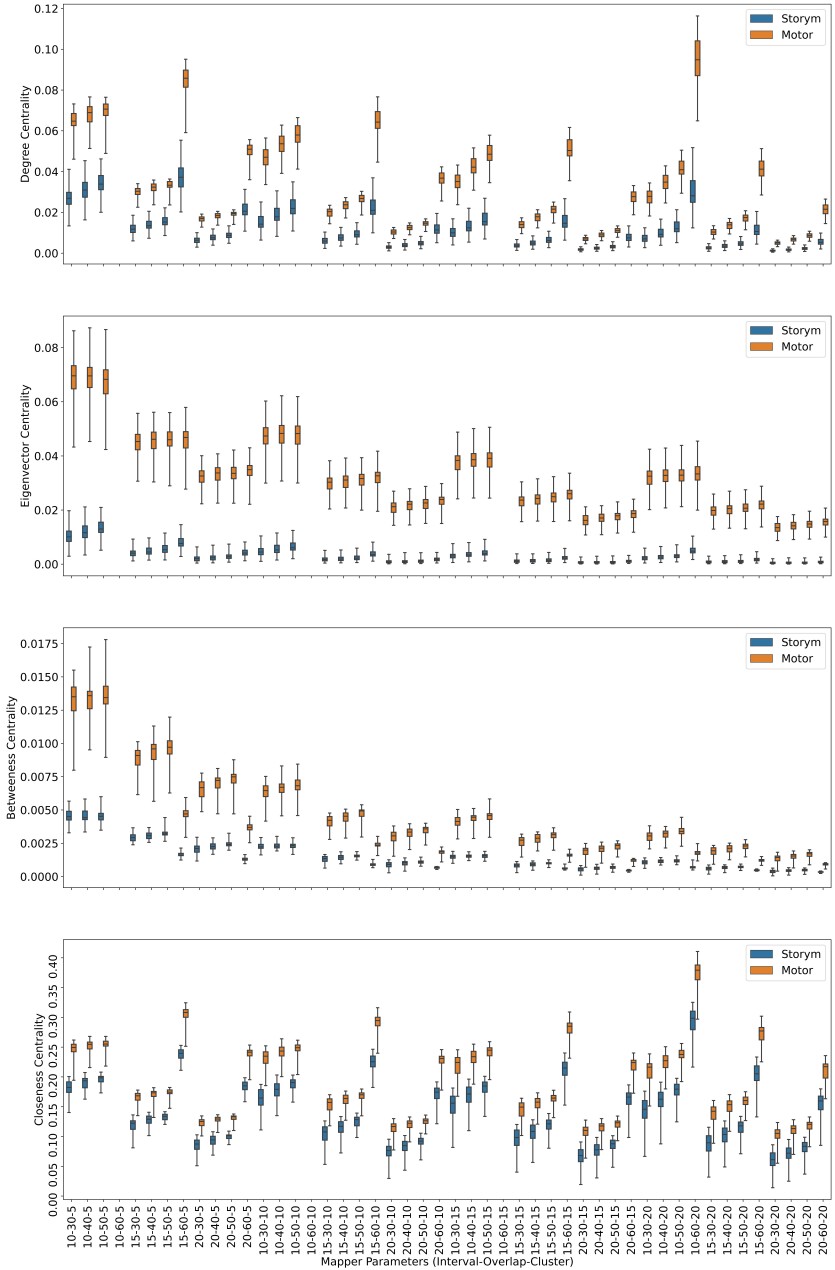

**Figure 4** **Story/math *vs.* Sensory-motor.** Box plots showing (from top to bottom) the degree, eigenvector, betweenness, and closeness centrality scores of the nodes of the Mapper graphs under different parameters. Parameters $(10-60-5), (10-60-10), (10-60-15)$ are missing due to high complexity of the calculations and the limitations of our workstation.

## Analysis pipeline

We summarize all the steps explained so far in the previous sections in Fig. 5.

Earlier studies (*Saggar et al., 2018*; *Saggar & Uddin, 2019*; *Duman, Tatar & Pirim, 2019*) show that the topological properties of Mapper graphs are robust by construction to
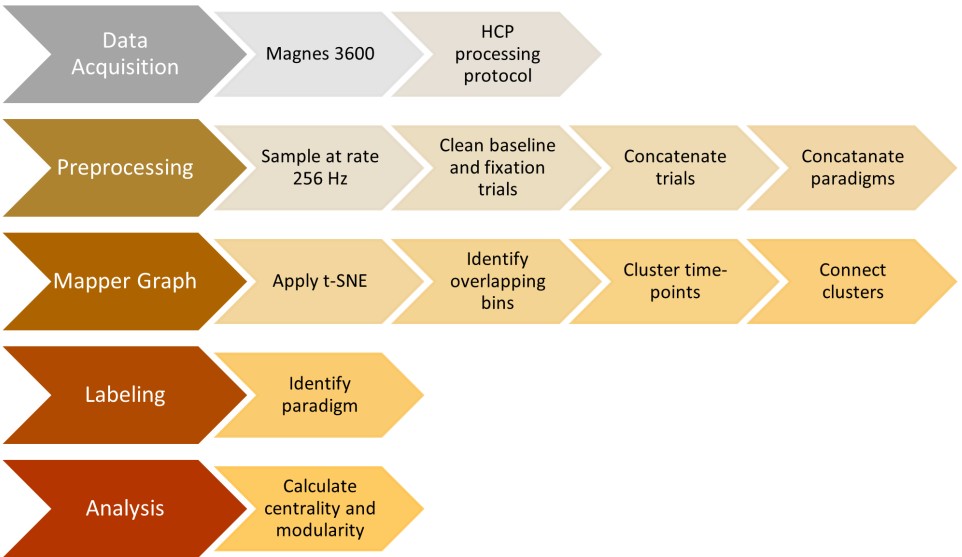

**Figure 5** **Steps of the proposed analysis.** Data collection, data preprocessing, mapper graph, labeling, centrality and community analysis.

parameter perturbations. To ensure the reliability of the statistical results, we tested our null hypothesis using 48 different sets of Mapper parameters. The domains for # intervals, overlap %, and # clusters are $\{10, 15, 20\}$, $\{30, 40, 50, 60\}$, and $\{5, 10, 15, 20\}$, respectively. The parameter space is chosen in a way that the resulting Mapper graphs are connected to make modularity and centrality calculation possible. Statistically significant and reliable results are obtained for the large portion of parameter values (see Results for details).

To have a better understanding, we also track the steps of the proposed analysis with the story/math and sensory-motor paradigm time points of the subject, 106521.

After concatenating the paradigms across the common channels, the subject, 106521, is represented by a matrix of dimensions $(357, 371 \times 232)$. The first 177,162 time points of the 357,371 time points belong to the story/math paradigm and the rest to the motor-sensory. The Mapper algorithm with the projected data by SNE and the parameters 10, 50, and 10 corresponding to the number of intervals, percentage of overlap, and the number of clusters in the single linkage clustering, respectively, outputs the Mapper graph in Fig. 6. It has 960 nodes and 13,889 edges. The nodes are either in black if the majority of the time points in that node belong to the story/math paradigm or in green color if the majority of the time points in that node belong to the sensory-motor paradigm. In 70% of the nodes the story/math paradigm time points are in majority and these nodes are on the periphery of the Mapper graph. The remaining 30% of the nodes are dominated by the sensory-motor paradigm time points which are placed mostly in the center of the graph.

## Code accessibility

The code described in the article is available from the corresponding author upon reasonable request.

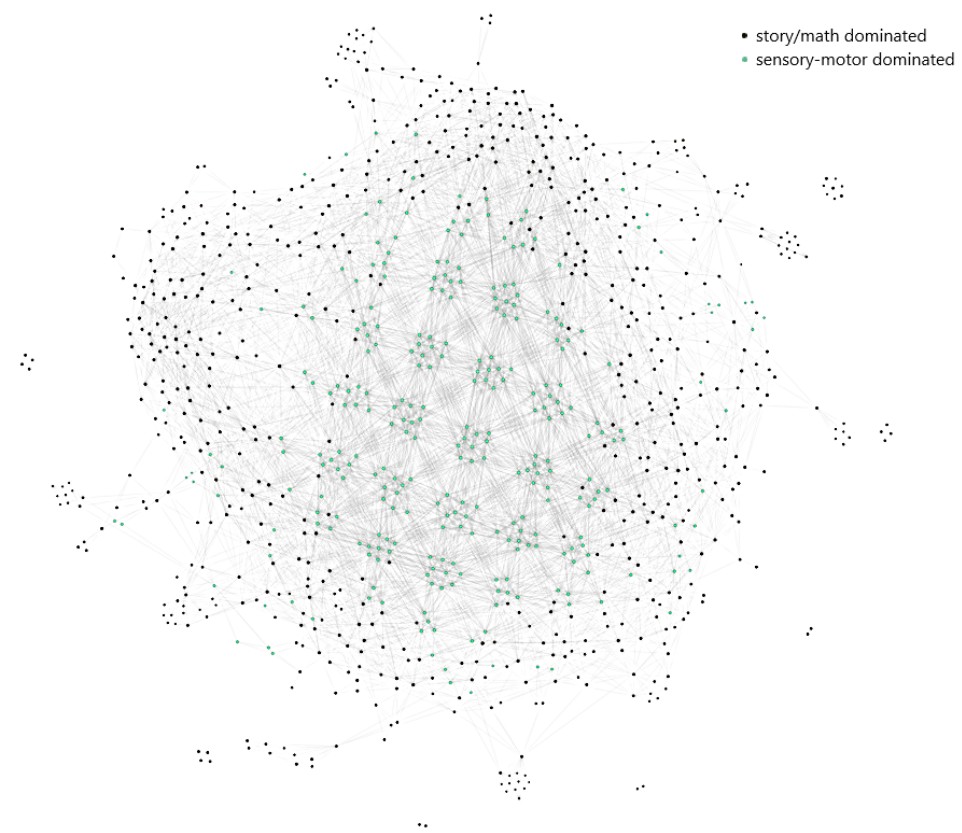

**Figure 6** **The Mapper graph of the subject 106521.** Nodes contain story/math and sensory-motor time points. Black nodes have story/math time points in majority whereas green nodes are dominated by the sensory-motor time points.

## RESULTS

In this section, we discuss the results of the experiments whose details are explained in the analysis pipeline. We check the reliability of our results by repeating the experiments with 48 different sets of Mapper parameters. The experiments are carried out in a Python environment using a workstation with 2 Nvidia GPUs (RTX 2080 Ti, 11 GB VRAM per GPU), 10 cores CPU (Intel i9-9820X), and 64 GB memory.

### Centrality of the Mapper graphs

For every pairwise scenario, working memory *vs.* story/math, working memory *vs.* sensory-motor, and story/math *vs.* sensory-motor and for every subject, we calculate four different centrality scores, degree, eigenvector, betweenness, and closeness centralities of the nodes in the Mapper graphs. Moreover, to show that the results do not depend on the Mapper graph, we repeat the same experiment with 48 different sets of Mapper parameters. The Mapper parameter space is made of triples of the form (# intervals, overlap %, # clusters). The domains for # intervals, overlap %, and # clusters are $\{10, 15, 20\}$, $\{30, 40, 50, 60\}$, and $\{5, 10, 15, 20\}$, respectively.

We look at the pairwise results. In the working memory and story/math paradigms, there are 60 common subjects. Figure 2 shows that, regardless of the Mapper parameters, all four centrality scores clearly distinguish the nodes dominated by the working memory time points from the nodes dominated by the story/math time points. We want to confirm this visual difference through statistical tests. The most relevant statistical test to use is the paired $t$-test that compares the mean of the centrality scores of the nodes dominated by the working memory time-points ($\mu_W$) and the mean of the centrality scores of the nodes dominated by the story/math paradigm's time-points ($\mu_S$). However, the paired $t$-test only works under the assumption that sample points follow a normal distribution. Therefore, we shall check numerically using the Shapiro–Wilk normality test or visually using probability plots that the differences in centrality scores of different paradigms satisfy this fact.

There might be Mapper plots with the differences in the centrality scores not following a normal distribution. For example, the distribution of closeness centrality scores of the nodes of the Mapper graph with parameters 10-40-5 is non normal distribution with $p = 0.02$ Shapiro Wilk normality test (see Fig. 7 for the probability plot). For such cases, we use the paired permutation test (*Good, 2013*) with the test statistic being the mean of the differences between the centrality scores. We shall clarify how the $p$ value of the test statistics is calculated. First, we calculate the observed mean differences $\mu_{\mathrm{obs}}$ between the working memory and the story/math paradigms. The permutation test assumes that there is no difference between the test statistics. The translation of this null hypothesis to our case is that there is no difference between the paradigms. Hence, we can create new data sets by swapping the centrality measures of the paradigms for the same subject. We compute the mean of the differences between the paradigms for every possible new data set and compare them to $\mu_{\mathrm{obs}}$. If the alternative hypothesis is that the means are not equal, then the $p$ value is $2\min(p_l, p_g)$ where $p_g$ (resp. $p_l$) is the probability of the mean of the differences being greater (resp. less) than $\mu_{\mathrm{obs}}$. If the alternative hypothesis is that one of the means is greater (resp. less) than the other mean, then the $p$ is the probability of the mean of the differences being greater (resp. less) than $\mu_{\mathrm{obs}}$. As in the other hypothesis testing, if the $p$ value is smaller than the significance level of 0.05, we say that the null hypothesis is not likely to be observed and favor the alternative. In Table 1, we illustrate the permutation test on the closeness centrality scores of the Mapper graph with parameters 10-40-5. We show how the paired permutation test creates a permuted data set. Later, the test statistics which is the mean of the differences in the centrality scores of the paradigms working memory and story/math are calculated. Going back to hypothesis testing, to verify our claims, we start with the null hypothesis:

$H_0$:  The mean of the centrality scores of the working memory nodes is equal to the mean of the centrality scores of the story/math nodes.

Based on the Figs. 2, 3 and 4, we expect $H_0$ to be rejected for all the Mapper parameters and for all centrality score types. Then, we verify which mean is greater by simply comparing the sample means. We look at the pairwise combinations one by one. To our expectations, in working memory and story/math combination, the $p$ values in the columns $\mu_W = \mu_S$ of the Table 2 being less than 0.05 indicate that we shall reject the null hypothesis in favor of

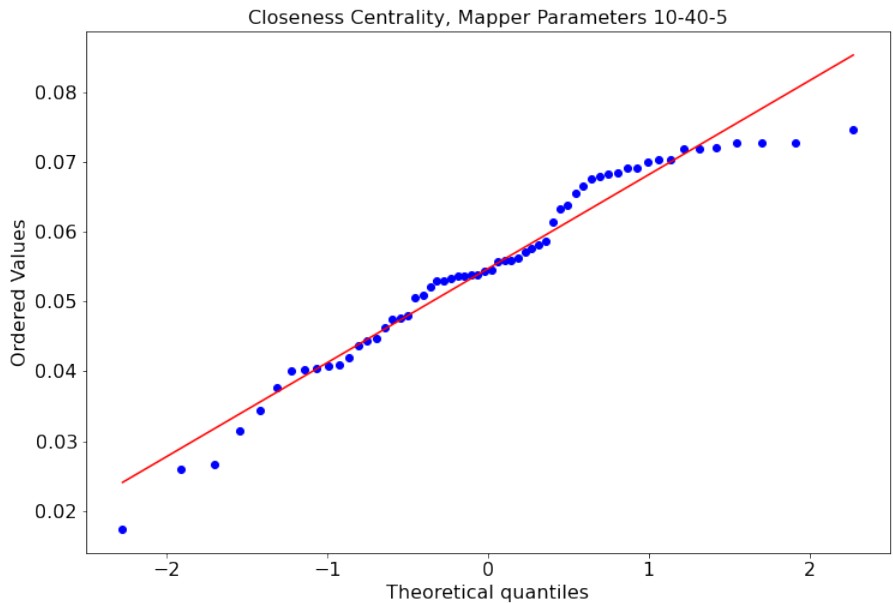

**Figure 7** **Non-normal distribution ($p = 0.02$ Shapiro–Wilk normality test) of the closeness centrality scores of the nodes of the Mapper graph with parameters 10-40-5.**

its alternative which is that $\mu_W$ and $\mu_S$ are significantly different. Moreover, as $\mu_S < \mu_W$ and halves of all the $p$ values in Table 2 are also less than 0.05, we can deduce that the mean of the centrality scores of the working memory nodes $\mu_W$ is significantly greater than the mean of the centrality scores of the story/math nodes $\mu_S$.

In the case of story/math and sensory-motor combination with 43 common subjects, by going through similar steps as above, we come to the conclusion using the test results in Table 3, that the mean of the centrality scores of the sensory-motor nodes $\mu_M$ is significantly greater than the mean of the centrality scores of the story/math nodes $\mu_S$.

For the final combination of working memory and sensory-motor with 21 common subjects, the Table 4 shows that the $p$ values of either the paired $t$-test or the permutation test under the null hypothesis are all greater than the significance level of 0.05 indicating that there is not enough evidence to reject the null hypothesis. Hence, we conclude for all types of centrality scores that mean the mean of the centrality scores of the working memory $\mu_W$ and sensory-motor $\mu_M$ are not significantly different.

The above discussions show that, when compared pairwise, the centrality score of the nodes dominated by the time points of the story/math paradigm is the smallest. We summarize our findings in the Table 5.

## Community structure and response time

Subjects' performances during the working memory and story/math experimental paradigms are scored and timed. In this section, we investigate if the modularity structure of the Mapper graphs is related to these measurements.

**Table 1** **Permutation Test.** Bold rows represent the swapped centrality scores to create a permutation of the observed closeness centrality scores of the Mapper graph with parameters 10-40-5. The permutation test compares the mean of the differences (Wrkmem–Storym) of the permuted centrality scores to the mean of the differences of the observed centrality scores which are 0.0490 and 0.0547, respectively.

| | Observed | | Permuted | |
|---|---|---|---|---|
| **Subjects** | **Wrkmem** | **Storym** | **Wrkmem** | **Storym** |
| 100307 | 0.2563 | 0.2006 | 0.2563 | 0.2006 |
| 102816 | 0.2372 | 0.1864 | 0.2372 | 0.1864 |
| 104012 | 0.2388 | 0.2045 | 0.2388 | 0.2045 |
| 106521 | 0.2301 | 0.1881 | 0.2301 | 0.1881 |
| 108323 | 0.2477 | 0.175 | 0.2477 | 0.175 |
| 109123 | 0.259 | 0.1862 | 0.259 | 0.1862 |
| 116726 | 0.2235 | 0.1968 | 0.2235 | 0.1968 |
| 133019 | 0.2497 | 0.1992 | 0.2497 | 0.1992 |
| 140117 | 0.2349 | 0.1942 | 0.2349 | 0.1942 |
| 146129 | 0.251 | 0.1871 | 0.251 | 0.1871 |
| 149741 | 0.2653 | 0.1967 | 0.2653 | 0.1967 |
| 151526 | 0.2534 | 0.1843 | 0.2534 | 0.1843 |
| 156334 | 0.2549 | 0.1963 | 0.2549 | 0.1963 |
| 158136 | 0.2528 | 0.1914 | 0.2528 | 0.1914 |
| 162026 | 0.2557 | 0.1981 | 0.2557 | 0.1981 |
| 166438 | 0.2579 | 0.2036 | 0.2579 | 0.2036 |
| 169040 | 0.2604 | 0.1883 | **0.1883** | **0.2604** |
| 175540 | 0.2364 | 0.1889 | 0.2364 | 0.1889 |
| 182840 | 0.2549 | 0.187 | 0.2549 | 0.187 |
| 185442 | 0.238 | 0.1748 | 0.238 | 0.1748 |
| 191033 | 0.2458 | 0.2021 | 0.2458 | 0.2021 |
| 191437 | 0.2373 | 0.1814 | 0.2373 | 0.1814 |
| 192641 | 0.2477 | 0.1749 | 0.2477 | 0.1749 |
| 195041 | 0.2562 | 0.1878 | 0.2562 | 0.1878 |
| 200109 | 0.2378 | 0.2118 | 0.2378 | 0.2118 |
| 204521 | 0.2449 | 0.1794 | 0.2449 | 0.1794 |
| 205119 | 0.2504 | 0.197 | 0.2504 | 0.197 |
| 212318 | 0.2413 | 0.1867 | 0.2413 | 0.1867 |
| 212823 | 0.2454 | 0.2006 | 0.2454 | 0.2006 |
| 214524 | 0.2452 | 0.2049 | 0.2452 | 0.2049 |
| 223929 | 0.2289 | 0.2115 | 0.2289 | 0.2115 |
| 248339 | 0.2403 | 0.1728 | 0.2403 | 0.1728 |
| 255639 | 0.246 | 0.1714 | 0.246 | 0.1714 |
| 257845 | 0.2565 | 0.2003 | **0.2003** | **0.2565** |
| 283543 | 0.2675 | 0.201 | 0.2675 | 0.201 |
| 293748 | 0.2587 | 0.1883 | 0.2587 | 0.1883 |
| 353740 | 0.2471 | 0.1941 | 0.2471 | 0.1941 |

**Table 1** (*continued*)

| Subjects | Observed | | Permuted | |
|---|---|---|---|---|
| | **Wrkmem** | **Storym** | **Wrkmem** | **Storym** |
| 433839 | 0.2485 | 0.1766 | 0.2485 | 0.1766 |
| 512835 | 0.2546 | 0.1846 | 0.2546 | 0.1846 |
| 555348 | 0.2528 | 0.1824 | 0.2528 | 0.1824 |
| 568963 | 0.2376 | 0.1933 | **0.1933** | **0.2376** |
| 599671 | 0.2382 | 0.1812 | 0.2382 | 0.1812 |
| 601127 | 0.2503 | 0.1965 | 0.2503 | 0.1965 |
| 660951 | 0.2488 | 0.2012 | 0.2488 | 0.2012 |
| 662551 | 0.2379 | 0.1976 | 0.2379 | 0.1976 |
| 665254 | 0.2509 | 0.1817 | 0.2509 | 0.1817 |
| 667056 | 0.2392 | 0.1856 | 0.2392 | 0.1856 |
| 679770 | 0.2308 | 0.1993 | 0.2308 | 0.1993 |
| 706040 | 0.2086 | 0.1677 | 0.2086 | 0.1677 |
| 707749 | 0.2122 | 0.166 | 0.2122 | 0.166 |
| 715950 | 0.2473 | 0.1914 | 0.2473 | 0.1914 |
| 725751 | 0.2648 | 0.1928 | 0.2648 | 0.1928 |
| 735148 | 0.2398 | 0.2021 | 0.2398 | 0.2021 |
| 783462 | 0.2479 | 0.1897 | 0.2479 | 0.1897 |
| 814649 | 0.2379 | 0.1899 | 0.2379 | 0.1899 |
| 825048 | 0.2393 | 0.1993 | 0.2393 | 0.1993 |
| 872764 | 0.2388 | 0.1849 | 0.2388 | 0.1849 |
| 877168 | 0.2377 | 0.1856 | 0.2377 | 0.1856 |
| 891667 | 0.2473 | 0.1943 | 0.2473 | 0.1943 |
| 917255 | 0.2564 | 0.2027 | 0.2564 | 0.2027 |

The distribution of modularity scores of all subjects are shown by box plots for different set of mapper parameters in Fig. 8. $x$-axis of the figure represents different set of mapper parameter. The interval parameter given by the first two numbers are either 10, 15 or 20. The following two numbers for the cluster parameter are either 30, 40, 50 or 60. The remaining numbers represents the overlap percentage which is either 5, 10, 15 or 20. Once we give a closer look at Fig. 8, we observe that the median of the modularity scores tends to

1. increase with the increase in the interval parameter as an increase in that parameter results in more nodes, hence more edges in the Mapper graph (see Fig. 9),
2. decrease with the increase in cluster parameter as an increase in that parameter results in a decrease in the number of edges (see Fig. 10),
3. non-decrease with the increase in the overlap percentage parameter as an increase in that parameter up to certain level results in a high number of edges and beyond that level less number of edges since some of the small nodes is absorbed by the larger ones. According to Fig. 11, that level is between 50% and 60% for our data set.

Moreover, the experiment results show that the modularity score of the Mapper graphs is negatively correlated to the reaction time. According to Table 6, the negative correlation is observed in all the Mapper graphs with different parameters. All correlation scores,

**Table 2  Working memory *vs.* Story/math.** Bonferroni corrected *p* values of either the paired *t*-test or the paired permutation test for all types of centrality scores. Bolded *p* values in every column represent the maximum of that column. Note that maximum values in all columns are less than the significance level of 0.05 and so are their halves.

| Parameter | Degree centrality | | Eigenvector centrality | | Betweenness centrality | | Closeness centrality | |
|---|---|---|---|---|---|---|---|---|
| (I-O-C) | $\mu_W = \mu_S$ | Test type | $\mu_W = \mu_S$ | Test type | $\mu_W = \mu_S$ | Test type | $\mu_W = \mu_S$ | Test type |
| 10305 | **0.009** | (P) | 1.34E−39 | (T) | 4.23E−42 | (T) | 1.49E−36 | (T) |
| 10405 | 3.87E−33 | (T) | 8.91E−38 | (T) | 2.64E−37 | (T) | **0.009** | (P) |
| 10505 | 8.37E−34 | (T) | 1.22E−37 | (T) | 1.68E−38 | (T) | 1.64E−38 | (T) |
| 15305 | 4.28E−34 | (T) | 3.40E−43 | (T) | 4.07E−41 | (T) | 2.08E−31 | (T) |
| 15405 | 3.27E−33 | (T) | 1.84E−42 | (T) | 1.43E−40 | (T) | 4.68E−35 | (T) |
| 15505 | 9.31E−33 | (T) | 3.26E−41 | (T) | 1.20E−39 | (T) | 2.11E−37 | (T) |
| 15605 | 1.52E−35 | (T) | 9.00E−39 | (T) | 1.85E−41 | (T) | 3.24E−38 | (T) |
| 20305 | 1.30E−34 | (T) | 3.70E−45 | (T) | 8.55E−41 | (T) | 7.20E−29 | (T) |
| 20405 | 3.54E−33 | (T) | 5.49E−45 | (T) | 2.00E−42 | (T) | 2.09E−31 | (T) |
| 20505 | 1.58E−31 | (T) | 1.51E−43 | (T) | 4.77E−41 | (T) | 1.31E−34 | (T) |
| 20605 | 6.88E−35 | (T) | 1.30E−41 | (T) | 3.94E−42 | (T) | 7.34E−38 | (T) |
| 103010 | 2.14E−38 | (T) | 2.71E−42 | (T) | 5.13E−45 | (T) | 1.40E−32 | (T) |
| 104010 | 2.96E−35 | (T) | 2.31E−39 | (T) | 5.54E−40 | (T) | 1.18E−33 | (T) |
| 105010 | 1.32E−35 | (T) | 8.55E−39 | (T) | 1.30E−39 | (T) | 4.40E−36 | (T) |
| 106010 | 5.00E−39 | (T) | **2.56E−36** | (T) | 1.58E−42 | (T) | 1.66E−38 | (T) |
| 153010 | 3.02E−36 | (T) | 3.18E−45 | (T) | 1.49E−38 | (T) | 1.56E−28 | (T) |
| 154010 | 9.81E−36 | (T) | 2.94E−44 | (T) | 2.29E−41 | (T) | 3.03E−29 | (T) |
| 155010 | 1.15E−35 | (T) | 6.53E−44 | (T) | 1.80E−42 | (T) | 2.46E−33 | (T) |
| 156010 | 6.66E−37 | (T) | 1.85E−40 | (T) | 1.31E−42 | (T) | 7.83E−36 | (T) |
| 203010 | 2.93E−35 | (T) | 3.81E−46 | (T) | **0.009** | (P) | 8.87E−27 | (T) |
| 204010 | 3.28E−35 | (T) | 4.43E−46 | (T) | **0.009** | (P) | 2.32E−27 | (T) |
| 205010 | 1.10E−34 | (T) | 9.81E−46 | (T) | **0.009** | (P) | 4.72E−30 | (T) |
| 206010 | 1.32E−36 | (T) | 4.41E−44 | (T) | 1.43E−44 | (T) | 1.58E−33 | (T) |
| 103015 | 6.66E−39 | (T) | 5.22E−44 | (T) | 2.30E−43 | (T) | 3.70E−31 | (T) |
| 104015 | 1.89E−36 | (T) | 5.35E−41 | (T) | 1.51E−42 | (T) | 5.81E−31 | (T) |
| 105015 | 1.53E−36 | (T) | 1.20E−39 | (T) | 2.62E−39 | (T) | 3.43E−34 | (T) |
| 153015 | 2.78E−36 | (T) | 1.24E−45 | (T) | 3.01E−37 | (T) | 6.75E−28 | (T) |
| 154015 | 3.49E−36 | (T) | 3.40E−45 | (T) | 3.87E−39 | (T) | 2.25E−27 | (T) |
| 155015 | 1.43E−36 | (T) | 5.58E−45 | (T) | 1.36E−43 | (T) | 2.61E−31 | (T) |
| 156015 | 2.11E−37 | (T) | 1.19E−41 | (T) | 1.34E−43 | (T) | 5.85E−34 | (T) |
| 203015 | 4.21E−35 | (T) | 3.94E−46 | (T) | 7.42E−34 | (T) | 2.08E−26 | (T) |
| 204015 | 2.44E−35 | (T) | 2.77E−46 | (T) | 2.21E−36 | (T) | 7.02E−26 | (T) |
| 205015 | 3.76E−35 | (T) | 4.16E−46 | (T) | 6.88E−40 | (T) | 2.78E−27 | (T) |
| 206015 | 4.64E−37 | (T) | 4.29E−45 | (T) | 1.16E−44 | (T) | 6.35E−31 | (T) |
| 103020 | 9.99E−39 | (T) | 1.40E−44 | (T) | 2.61E−42 | (T) | 1.40E−44 | (T) |
| 104020 | 8.23E−37 | (T) | 7.69E−42 | (T) | 8.51E−42 | (T) | 7.69E−42 | (T) |
| 105020 | 2.20E−37 | (T) | 5.58E−41 | (T) | 3.94E−41 | (T) | 5.58E−41 | (T) |
| 153020 | 1.00E−35 | (T) | 1.39E−45 | (T) | 3.68E−35 | (T) | 1.39E−45 | (T) |
| 154020 | 4.86E−36 | (T) | 1.69E−45 | (T) | 4.59E−38 | (T) | 1.69E−45 | (T) |

**Table 2** (*continued*)

| Parameter | Degree centrality | | Eigenvector centrality | | Betweenness centrality | | Closeness centrality | |
|---|---|---|---|---|---|---|---|---|
| (I-O-C) | $\mu_W = \mu_S$ | Test type | $\mu_W = \mu_S$ | Test type | $\mu_W = \mu_S$ | Test type | $\mu_W = \mu_S$ | Test type |
| 155020 | 1.06E−36 | (T) | 4.68E−45 | (T) | 2.05E−42 | (T) | 4.68E−45 | (T) |
| 156020 | 8.73E−38 | (T) | 1.25E−42 | (T) | 8.42E−44 | (T) | 1.25E−42 | (T) |
| 203020 | 1.09E−34 | (T) | 1.09E−45 | (T) | 5.00E−32 | (T) | 1.09E−45 | (T) |
| 204020 | 5.80E−35 | (T) | 9.05E−46 | (T) | 1.17E−34 | (T) | 9.05E−46 | (T) |
| 205020 | 4.99E−35 | (T) | 4.64E−46 | (T) | 4.20E−38 | (T) | 4.64E−46 | (T) |
| 206020 | 4.43E−37 | (T) | 1.16E−45 | (T) | 3.16E−44 | (T) | 1.16E−45 | (T) |

with the strongest recorded at the parameters $(10 - 40 - 5)$ (see Fig. 12), are weak and non-significant.

We also note that the strongest correlation scores are associated with the lowest clustering parameters as an increase in this parameter decreases the number of edges within the same interval which more likely contains similar tasks (see Fig. 13).

### Other methods

We also visualize the time points before the Mapper algorithm. In Fig. 14, we see that the story/math paradigm time points overlap with the sensory-motor paradigm time points which indicates that a distance-based clustering would not distinguish them.

## DISCUSSION

The neural oscillations in the brain change rapidly in response to sensory and cognitive stimulations (*Pfurtscheller & Lopes da Silva, 1999*). As this fact implies the quick change in functional connectivity patterns, it is crucial to analyze dynamic connectivity to gain insight into how the information is processed in the brain. There is also increasing clinical interest in dynamic functional connectivity, which is shown to be perturbed by diseases such as schizophrenia (*Damaraju et al., 2014*), bipolar disorder (*Rashid et al., 2014*), and depression (*Demirtaş et al., 2016*). For clinical applications, there is a need to have methods that derive a meaningful conclusion from high spatiotemporal dimensional data on the individual level. Moreover, state-of-the-art methods require the temporal and spatial collapse of the data, which may result in information loss. *Saggar et al. (2018)* has recently addressed these issues for fMRI data using the Mapper algorithm that provides an interactive simple visualization of the brain dynamics on an individual level during ongoing cognitive tasks. However, it is hard to detect the rapid changes in neural activity using hemodynamic signals from fMRI, as they are a proxy of neural activity. In this current work, we extend this topological approach to high temporal resolution MEG data which can measure the fast fluctuations directly avoiding the autocorrelation structure caused by the hemodynamic response in fMRI.

This new topological approach to the MEG data provides an interactive mathematical graph that tracks the brain configuration patterns of each participant during the sensory-motor, story/math, and working memory tasks. The Mapper graph for each participant is obtained from pairwise concatenation of the MEG data sets of different tasks which are not

**Table 3 Story/math *vs.* Sensory-motor.** Bonferroni corrected *p* values of either the paired *t*-test or the paired permutation test for all types of centrality scores. Bolded *p* values in every column represent the minimum of that column. Note that the minimum values in all columns are less than the significance level of 0.05 and so are their halves.

| Parameter | Degree centrality | | Eigenvector centrality | | Betweenness centrality | | Closeness centrality | |
|---|---|---|---|---|---|---|---|---|
| (I-O-C) | $\mu_S = \mu_M$ | Test type | $\mu_S = \mu_M$ | Test type | $\mu_S = \mu_M$ | Test type | $\mu_S = \mu_M$ | Test type |
| 10305 | **0.009** | (P) | 1.78E−31 | (T) | **0.009** | (P) | **0.009** | (P) |
| 10405 | **0.009** | (P) | 4.91E−31 | (T) | **0.009** | (P) | **0.009** | (P) |
| 10505 | **0.009** | (P) | 3.58E−29 | (T) | **0.009** | (P) | **0.009** | (P) |
| 15305 | **0.009** | (P) | 1.84E−34 | (T) | **0.009** | (P) | **0.009** | (P) |
| 15405 | **0.009** | (P) | 2.47E−33 | (T) | **0.009** | (P) | **0.009** | (P) |
| 15505 | **0.009** | (P) | **0.009** | (P) | **0.009** | (P) | **0.009** | (P) |
| 15605 | **0.009** | (P) | **0.009** | (P) | **0.009** | (P) | **0.009** | (P) |
| 20305 | **0.009** | (P) | 2.35E−35 | (T) | **0.009** | (P) | **0.009** | (P) |
| 20405 | **0.009** | (P) | **0.009** | (P) | **0.009** | (P) | **0.009** | (P) |
| 20505 | **0.009** | (P) | **0.009** | (P) | **0.009** | (P) | **0.009** | (P) |
| 20605 | **0.009** | (P) | 9.49E−33 | (T) | **0.009** | (P) | **0.009** | (P) |
| 103010 | **0.009** | (P) | 1.12E−32 | (T) | **0.009** | (P) | **0.009** | (P) |
| 104010 | **0.009** | (P) | 8.19E−32 | (T) | **0.009** | (P) | **0.009** | (P) |
| 105010 | **0.009** | (P) | 1.62E−30 | (T) | **0.009** | (P) | **0.009** | (P) |
| 153010 | **0.009** | (P) | 8.73E−35 | (T) | **0.009** | (P) | **0.009** | (P) |
| 154010 | **0.009** | (P) | 8.37E−35 | (T) | **0.009** | (P) | **0.009** | (P) |
| 155010 | **0.009** | (P) | 1.24E−33 | (T) | **0.009** | (P) | **0.009** | (P) |
| 156010 | **0.009** | (P) | 2.83E−31 | (T) | **0.009** | (P) | **0.009** | (P) |
| 203010 | 3.87E−30 | (T) | 9.59E−35 | (T) | 3.8E−30 | (T) | **0.009** | (P) |
| 204010 | **0.009** | (P) | 5.49E−35 | (T) | **0.009** | (P) | **0.009** | (P) |
| 205010 | **0.009** | (P) | 6.84E−35 | (T) | **0.009** | (P) | **0.009** | (P) |
| 206010 | **0.009** | (P) | **0.009** | (P) | **0.009** | (P) | **0.009** | (P) |
| 103015 | **0.009** | (P) | 6.43E−33 | (T) | **0.009** | (P) | **0.009** | (P) |
| 104015 | **0.009** | (P) | 1.23E−32 | (T) | **0.009** | (P) | **0.009** | (P) |
| 105015 | **0.009** | (P) | 1.27E−30 | (T) | **0.009** | (P) | **0.009** | (P) |
| 153015 | 6.66E−30 | (T) | 1.11E−34 | (T) | **0.009** | (P) | **0.009** | (P) |
| 154015 | **0.009** | (P) | 1.04E−34 | (T) | **0.009** | (P) | **0.009** | (P) |
| 155015 | **0.009** | (P) | **0.009** | (P) | **0.009** | (P) | **0.009** | (P) |
| 156015 | **0.009** | (P) | 5.04E−32 | (T) | 9.72E−32 | (T) | **0.009** | (P) |
| 203015 | 8.1E−29 | (T) | 1.36E−33 | (T) | **0.009** | (P) | **0.009** | (P) |
| 204015 | 2.13E−29 | (T) | 8.14E−35 | (T) | 1.96E−30 | (T) | **0.009** | (P) |
| 205015 | **0.009** | (P) | 1.62E−34 | (T) | **0.009** | (P) | **0.009** | (P) |
| 206015 | **0.009** | (P) | **0.009** | (P) | **0.009** | (P) | **0.009** | (P) |
| 103020 | 2.62E−29 | (T) | 6.21E−33 | (T) | **0.009** | (P) | **0.009** | (P) |
| 104020 | **0.009** | (P) | 1.21E−32 | (T) | **0.009** | (P) | **0.009** | (P) |
| 105020 | **0.009** | (P) | 4.91E−31 | (T) | 1.55E−30 | (T) | **0.009** | (P) |
| 106020 | **0.009** | (P) | 1.68E−28 | (T) | 5.04E−28 | (T) | **0.009** | (P) |
| 153020 | 5.49E−29 | (T) | 2.43E−34 | (T) | **0.009** | (P) | **0.009** | (P) |
| 154020 | 1.23E−29 | (T) | 3.19E−34 | (T) | **0.009** | (P) | **0.009** | (P) |

**Table 3** (*continued*)

| Parameter | Degree centrality | | Eigenvector centrality | | Betweenness centrality | | Closeness centrality | |
|---|---|---|---|---|---|---|---|---|
| (I-O-C) | $\mu_S = \mu_M$ | Test type | $\mu_S = \mu_M$ | Test type | $\mu_S = \mu_M$ | Test type | $\mu_S = \mu_M$ | Test type |
| 155020 | **0.009** | (P) | **0.009** | (P) | **0.009** | (P) | **0.009** | (P) |
| 156020 | **0.009** | (P) | **0.009** | (P) | 1.44E−31 | (T) | **0.009** | (P) |
| 203020 | 6.39E−28 | (T) | 6.61E−33 | (T) | **0.009** | (P) | **0.009** | (P) |
| 204020 | 1.69E−28 | (T) | 8.91E−34 | (T) | 6.7E−29 | (T) | **0.009** | (P) |
| 205020 | 1.75E−29 | (T) | 6.03E−34 | (T) | **0.009** | (P) | **0.009** | (P) |
| 206020 | **0.009** | (P) | **0.009** | (P) | **0.009** | (P) | **0.009** | (P) |

[1]While the filter function t-SNE maps the time points to two dimensional space to make the visualization possible, it is still possible to trace back the activated locations in the brain from the nodes of the mapper graph (*Saggar et al., 2018*). Hence, the spatial information is not lost during the process.

temporally and spatially[1] collapsed in prior. The mesoscale graph invariants (*i.e.,* centrality and modularity) of the resulting graphs uncover temporal characteristics of the brain configuration. In line with the fMRI results (*Saggar et al., 2018*) the centrality invariants statistically differentiate story/math and working memory tasks . Working memory task being cognitively more demanding has greater centrality values than the story/math data set on the individual level. This result is in accordance with the earlier neurophysiological findings of brain dynamics (*Liu & Duyn, 2013*; *Ponce-Alvarez et al., 2015*) and previous fMRI results (*Saggar et al., 2018*) where the higher similarity between brain regions is observed if the task requires stronger cognitive involvement.

As noted by *O'Neill et al. (2018)*, there are two existing groups of approaches for aggregating the time points to measure connectivity between brain regions: (i) using multiple successive time points such as in the case of sliding windows,[2] and (ii) aggregating across the same time point of multiple trials to generate connectivity dynamics. The first type of approach is mainly used for experiments without any trials (*e.g.,* resting state), while the latter requires multiple task trials. The Mapper graph addresses the limitation of both approaches and can be used in the experiments both with or without trials. While there are parameters in the Mapper algorithm, the graphs are shown to be robust to the parameter choice. This is not the case with the sliding windows (or similar) methods, as the results with small window width are governed by the noise, while too large windows are not sensitive to rapid fluctuations. Unlike sliding windows, the temporal resolution is also preserved during the calculation of Mapper graphs. Moreover, the information loss caused by averaging over trials is avoided in the proposed Mapper pipeline.

The Mapper algorithm applied on high temporal resolution raw MEG data ($\sim$ 800,000 time points) also contributes to biological understanding. The following possible connectomic biomarker can be extracted from the Mapper graph: the centrality of the Mapper graph that distinguishes working memory, story/math, and sensory-motor tasks.

[2]Note that sliding windows method calculates region-by-region connectivity on each time window and then explores the change in standard FC patterns during the experiment.

The Mapper graphs with high modularity (*i.e.,* the nodes containing similar tasks have dense connections between them but sparse connections with the remaining nodes) have negative and non-significant correlation with the response time of the cognitive tasks. A stronger but statistically weak correlation between modularity and response time is also found in fMRI studies (*Saggar et al., 2018*). The weaker correlation in MEG is due to a high number of time points ($\sim$800,000) with higher noise compared to fMRI data, which has about 1,000 time points. In order to improve this results, the future studies can investigate

**Table 4  Working memory *vs.* Sensory-motor.** Bonferroni corrected *p* values of the paired *t*-test for all types of centrality scores. Bolded *p* values in every column represent the minimum of that column. Note that minimum values in all columns are greater than the significance level of 0.05.

| Parameter (I-O-C) | Degree Centrality $\mu_M = \mu_W$ | Test type | Eigenvector Centrality $\mu_M = \mu_W$ | Test type | Betweenness Centrality $\mu_M = \mu_W$ | Test type | Closeness Centrality $\mu_M = \mu_W$ | Test type |
|---|---|---|---|---|---|---|---|---|
| 10305 | 0.184 | (T) | 0.164 | (T) | 0.203 | (T) | 0.522 | (T) |
| 10405 | 0.198 | (T) | 0.167 | (T) | 0.204 | (T) | 0.396 | (T) |
| 10505 | 0.262 | (T) | 0.173 | (T) | 0.227 | (T) | 0.498 | (T) |
| 15305 | 0.126 | (T) | 0.134 | (T) | 0.165 | (T) | 0.248 | (T) |
| 15405 | 0.137 | (T) | 0.11 | (T) | 0.182 | (T) | 0.215 | (T) |
| 15505 | 0.165 | (T) | 0.167 | (T) | 0.174 | (T) | 0.288 | (T) |
| 15605 | 0.175 | (T) | 0.157 | (T) | 0.19 | (T) | 0.254 | (T) |
| 20305 | 0.0973 | (T) | 0.169 | (T) | 0.175 | (T) | 0.261 | (T) |
| 20405 | **0.0775** | (T) | 0.132 | (T) | 0.139 | (T) | 0.256 | (T) |
| 20505 | 0.0992 | (T) | 0.128 | (T) | 0.185 | (T) | 0.24 | (T) |
| 20605 | 0.156 | (T) | 0.129 | (T) | 0.202 | (T) | 0.34 | (T) |
| 103010 | 0.257 | (T) | 0.246 | (T) | 0.578 | (T) | 0.771 | (T) |
| 104010 | 0.231 | (T) | 0.227 | (T) | 0.447 | (T) | 0.686 | (T) |
| 105010 | 0.217 | (T) | 0.198 | (T) | 0.235 | (T) | 0.317 | (T) |
| 106010 | 0.387 | (T) | 0.305 | (T) | 0.869 | (T) | 0.611 | (T) |
| 153010 | 0.121 | (T) | 0.131 | (T) | 0.263 | (T) | 0.192 | (T) |
| 154010 | 0.126 | (T) | **0.106** | (T) | 0.177 | (T) | 0.331 | (T) |
| 155010 | 0.149 | (T) | 0.186 | (T) | 0.189 | (T) | 0.351 | (T) |
| 156010 | 0.167 | (T) | 0.163 | (T) | 0.282 | (T) | 0.235 | (T) |
| 203010 | 0.0884 | (T) | 0.122 | (T) | 0.136 | (T) | 0.147 | (T) |
| 204010 | 0.0921 | (T) | 0.12 | (T) | 0.159 | (T) | 0.227 | (T) |
| 205010 | 0.103 | (T) | 0.115 | (T) | 0.186 | (T) | 0.272 | (T) |
| 206010 | 0.131 | (T) | 0.137 | (T) | 0.188 | (T) | 0.261 | (T) |
| 103015 | 0.202 | (T) | 0.213 | (T) | 0.297 | (T) | 0.559 | (T) |
| 104015 | 0.205 | (T) | 0.188 | (T) | 0.408 | (T) | 0.578 | (T) |
| 105015 | 0.203 | (T) | 0.19 | (T) | 0.296 | (T) | 0.443 | (T) |
| 106015 | 0.335 | (T) | 0.298 | (T) | 0.667 | (T) | 0.606 | (T) |
| 153015 | 0.104 | (T) | 0.107 | (T) | 0.139 | (T) | **0.124** | (T) |
| 154015 | 0.112 | (T) | 0.117 | (T) | 0.153 | (T) | 0.307 | (T) |
| 155015 | 0.118 | (T) | 0.133 | (T) | 0.153 | (T) | 0.387 | (T) |
| 156015 | 0.131 | (T) | 0.148 | (T) | 0.136 | (T) | 0.331 | (T) |
| 203015 | 0.0893 | (T) | 0.14 | (T) | 0.101 | (T) | 0.13 | (T) |
| 204015 | 0.0926 | (T) | 0.138 | (T) | 0.112 | (T) | 0.165 | (T) |
| 205015 | 0.108 | (T) | 0.131 | (T) | 0.172 | (T) | 0.247 | (T) |
| 206015 | 0.124 | (T) | 0.13 | (T) | 0.122 | (T) | 0.232 | (T) |
| 103020 | 0.257 | (T) | 0.25 | (T) | 0.314 | (T) | 0.559 | (T) |
| 104020 | 0.199 | (T) | 0.201 | (T) | 0.368 | (T) | 0.63 | (T) |
| 105020 | 0.18 | (T) | 0.18 | (T) | 0.218 | (T) | 0.365 | (T) |
| 106020 | 0.265 | (T) | 0.24 | (T) | 0.484 | (T) | 0.498 | (T) |

**Table 4** (*continued*)

| Parameter (I-O-C) | Degree Centrality | | Eigenvector Centrality | | Betweenness Centrality | | Closeness Centrality | |
|---|---|---|---|---|---|---|---|---|
| | $\mu_M = \mu_W$ | Test type | $\mu_M = \mu_W$ | Test type | $\mu_M = \mu_W$ | Test type | $\mu_M = \mu_W$ | Test type |
| 153020 | 0.117 | (T) | 0.123 | (T) | **0.0964** | (T) | 0.164 | (T) |
| 154020 | 0.12 | (T) | 0.12 | (T) | 0.129 | (T) | 0.279 | (T) |
| 155020 | 0.106 | (T) | 0.115 | (T) | 0.119 | (T) | 0.374 | (T) |
| 156020 | 0.168 | (T) | 0.183 | (T) | 0.213 | (T) | 0.253 | (T) |
| 203020 | 0.0982 | (T) | 0.136 | (T) | 0.107 | (T) | 0.125 | (T) |
| 204020 | 0.101 | (T) | 0.137 | (T) | 0.125 | (T) | 0.141 | (T) |
| 205020 | 0.103 | (T) | 0.132 | (T) | 0.131 | (T) | 0.219 | (T) |
| 206020 | 0.131 | (T) | 0.148 | (T) | 0.136 | (T) | 0.331 | (T) |

**Table 5** **Summary of results.** Our assumptions on the centrality scores of the Mapper graphs of the paradigms working memory *vs.* story/math and story/math *vs.* sensory-motor are valid whereas it is not for the paradigms working memory *vs.* sensory-motor.

| Hypothesis | Result |
|---|---|
| $H_{WS}$ | Satisfied |
| $H_{WM}$ | Not Satisfied |
| $H_{SM}$ | Satisfied |

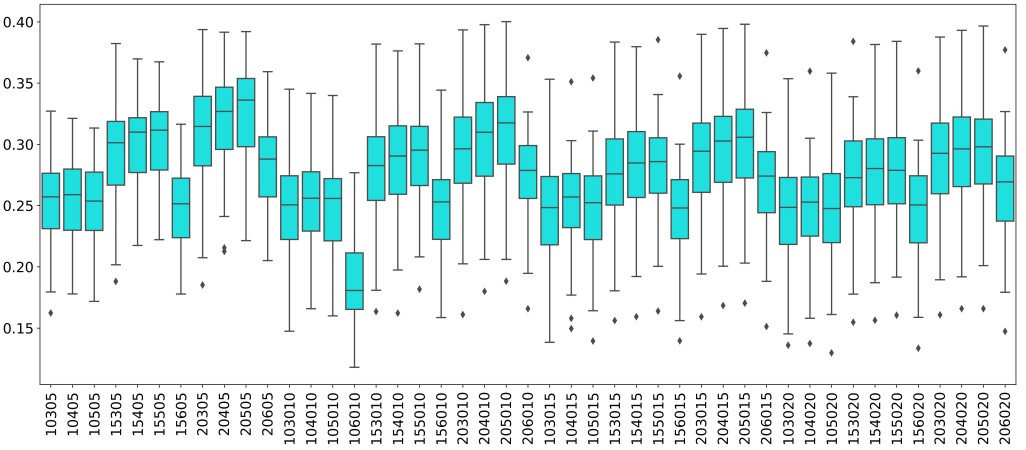

**Figure 8** **Modularity box plots.** The distribution of modularity scores of all subjects are given by box plots for each set of parameters. *x*-axis is for the mapper parameters. First two numbers (10, 15 or 20) are for the interval parameter, following two numbers (30,40,50 or 60) are for the cluster parameter and remaining numbers (5, 10, 15 or 20) are for the overlap percentage..

different filter functions other than t-SNE. Unlike fMRI, the proposed approach on MEG does not detect any correlation between accuracy and modularity possibly due to the high level of noise.

Another mesoscale structure that provides biological understanding is centrality. The four centrality invariants used in our analysis give similar outcomes for each participant's Mapper graphs. The high number of time points (∼400,000) per paradigm and the

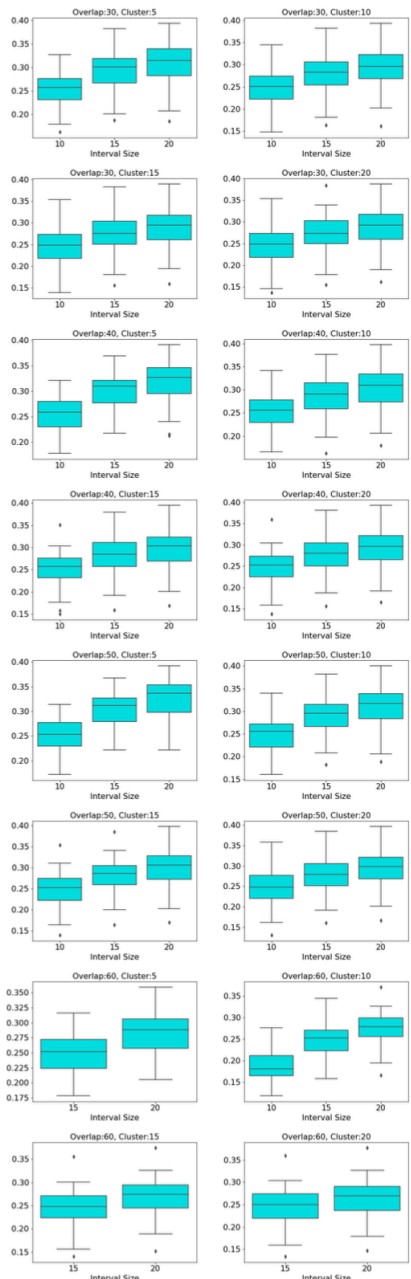

**Figure 9  Box plots across interval parameter.** Modularity box plots of Mapper graphs over changing interval parameter while the other parameters remain constant.

computational limitations necessitate the pairwise comparison of the working memory, story/math, and sensory-motor paradigms. The centrality of working memory and the sensory-motor tasks are shown to be greater than the story/math tasks due to higher cognitive demand, as a similar but statistically less significant result is also observed in fMRI (*Saggar et al., 2018*). This supports earlier findings indicating that the subjects performing tasks with higher cognitive efforts have higher similarity in whole-brain activation patterns

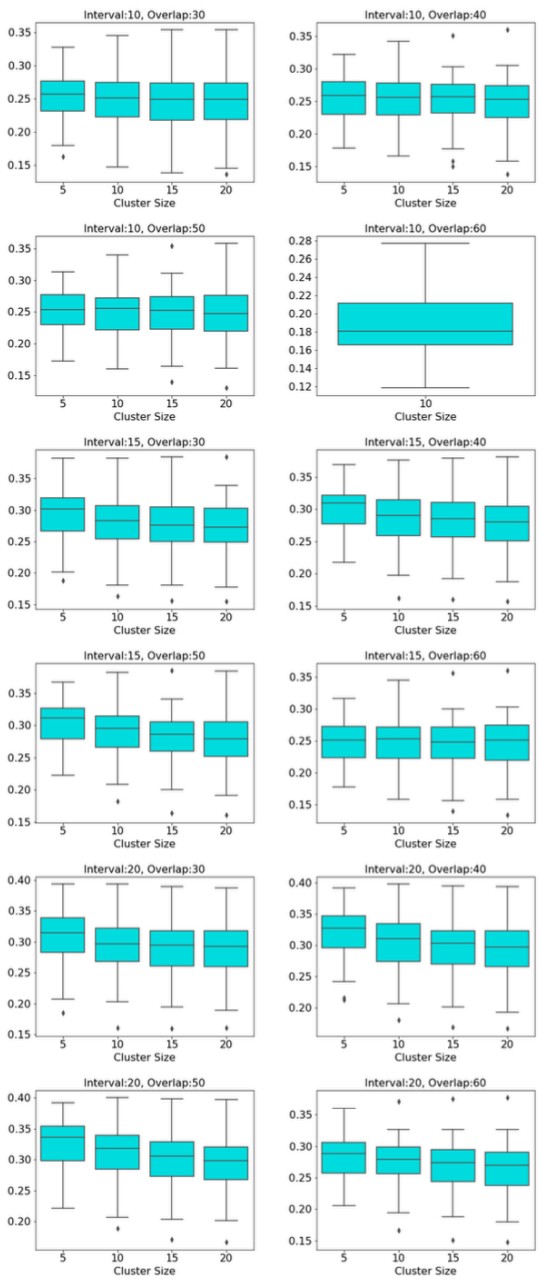

**Figure 10  Box plots across cluster parameter.** Modularity box plots of Mapper graphs over changing cluster parameter while the other parameters remain constant.

compared to the periods of rest (*Liu & Duyn, 2013*; *Ponce-Alvarez et al., 2015*). Thus, the nodes containing time points of cognitively demanding tasks have more edges connecting resulting in higher centrality scores than the nodes of less demanding tasks. In addition, no difference between the centrality of sensory-motor tasks and the centrality of working memory is observed. This result, which contradicts the fMRI results for the same tasks in *Saggar et al. (2018)*, can be explained by the low number of common subjects ($n = 21$) who

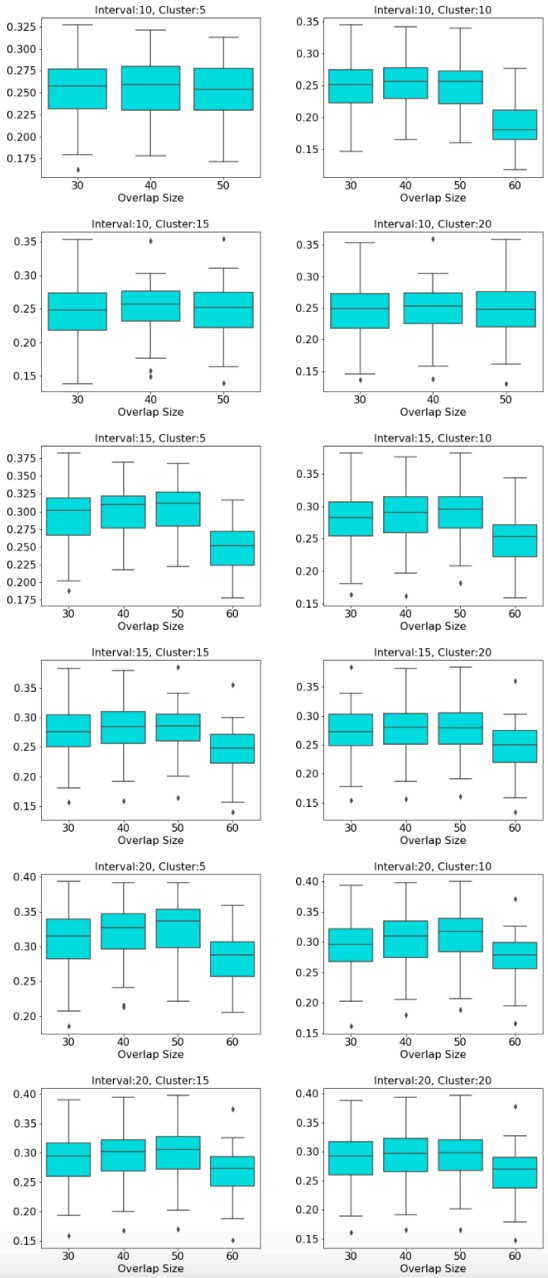

**Figure 11** **Box plots across overlap parameter.** Modularity box plots of Mapper graphs over changing overlap parameter while the other parameters remain constant.

performed both tasks. While a little fluctuation in the connectedness of specific brain areas during the motor-related tasks that were previously reported supports our findings (*Bassett et al., 2013*), the neurophysiological relationship between the motor and the cognitive skills is more complex and still under discussion. The proposed topological approach might shed light on this issue from the perspective of overall brain dynamics.

**Table 6 Modularity analysis.** Correlation scores in ascending order between the modularity scores and the reaction times for every Mapper parameter.

| Parameters | Correlation | Significance | Parameters | Correlation | Significance |
|---|---|---|---|---|---|
| 10405 | −0.267126 | 0.05557 | 204010 | −0.123708 | 0.382255 |
| 20505 | −0.260994 | 0.061645 | 104015 | −0.119964 | 0.396934 |
| 15505 | −0.260676 | 0.061974 | 153010 | −0.111893 | 0.429676 |
| 10505 | −0.25259 | 0.070826 | 156015 | −0.110886 | 0.433868 |
| 10305 | −0.238318 | 0.088873 | 103015 | −0.107187 | 0.44945 |
| 15405 | −0.234167 | 0.094747 | 206015 | −0.099824 | 0.48137 |
| 20605 | −0.222254 | 0.113281 | 104020 | −0.098844 | 0.485705 |
| 15305 | −0.217556 | 0.121308 | 205015 | −0.092262 | 0.515354 |
| 20405 | −0.215669 | 0.124651 | 156020 | −0.090867 | 0.521749 |
| 15605 | −0.20907 | 0.13689 | 154015 | −0.090226 | 0.524702 |
| 155010 | −0.179271 | 0.203498 | 155020 | −0.089571 | 0.52773 |
| 105010 | −0.177728 | 0.207481 | 103020 | −0.08614 | 0.543722 |
| 20305 | −0.174525 | 0.215921 | 206020 | −0.074192 | 0.601166 |
| 205010 | −0.168408 | 0.232699 | 203010 | −0.069045 | 0.626706 |
| 104010 | −0.167384 | 0.235594 | 204015 | −0.059161 | 0.676963 |
| 103010 | −0.157533 | 0.2647 | 153015 | −0.058248 | 0.681679 |
| 106010 | −0.142675 | 0.312971 | 205020 | −0.054341 | 0.702003 |
| 105015 | −0.141298 | 0.317711 | 153020 | −0.050865 | 0.720262 |
| 206010 | −0.140536 | 0.320354 | 154020 | −0.047769 | 0.736656 |
| 156010 | −0.134884 | 0.340397 | 204020 | −0.025744 | 0.856244 |
| 154010 | −0.133479 | 0.345494 | 203015 | −0.019687 | 0.889826 |
| 105020 | −0.131414 | 0.353076 | 203020 | 0.006811 | 0.96178 |
| 155015 | −0.127178 | 0.368944 | | | |

One of the limitations in this current study is the computational expense of generating a Mapper graph from 350,000–400,000 time points of the MEG recordings from each individual. It is worth noting that new software packages such as NeuMapper (*Geniesse, Chowdhury & Saggar, 2022*) are reported to be much more efficient compared to KeplerMapper which is used in the current study. Another limitation is to determine the minimum number of time points that are required to produce a robust generation of Mapper graphs and corresponding graph invariants. As it might not be feasible to acquire sufficiently long recordings from individuals for clinical studies. The other possible criticism might be the low number of subjects ($n = 21$) in the concatenated sensory-motor and working memory data which potentially results in higher adjusted $p$-values. The other pairs working memory *vs.* story/math ($n = 60$) and sensory-motor *vs.* story/math ($n = 45$) have a higher number of subjects where the results are more statistically significant. Even though it is shown that the results are robust to parameter variation, it will be useful to find a parameter and filter function selection and exploration framework for MEG data. In *Geniesse, Chowdhury & Saggar (2022)*, the authors propose an algorithm that leverages the autocorrelation structure present in fMRI data due to the slow hemodynamic response, which is not the case in MEG data. Another consideration is that results will be mainly

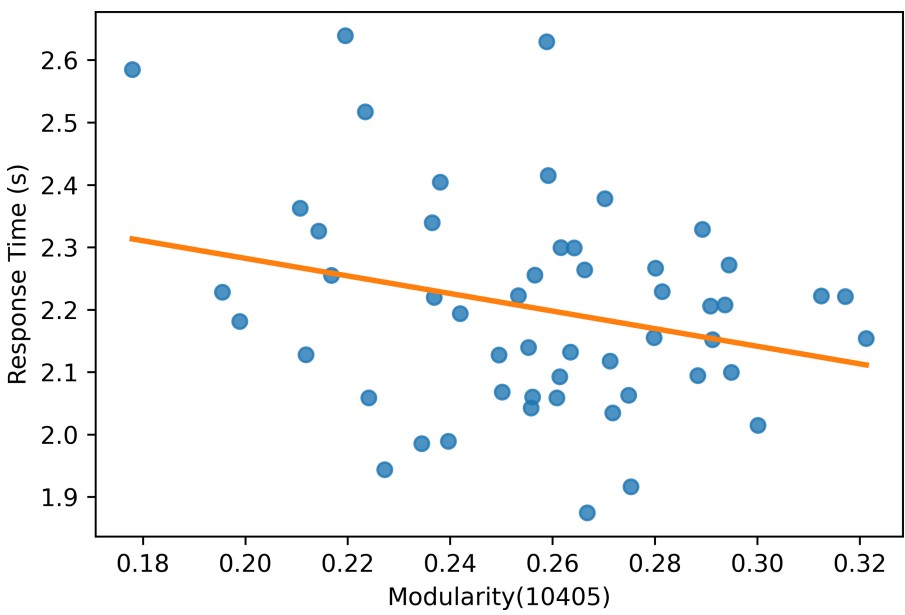

**Figure 12** **Modularity scatter plot.** Modularity scores of the Mapper graph with the parameters (10-40-5) *vs.* the response time in seconds with a fitted regression line.

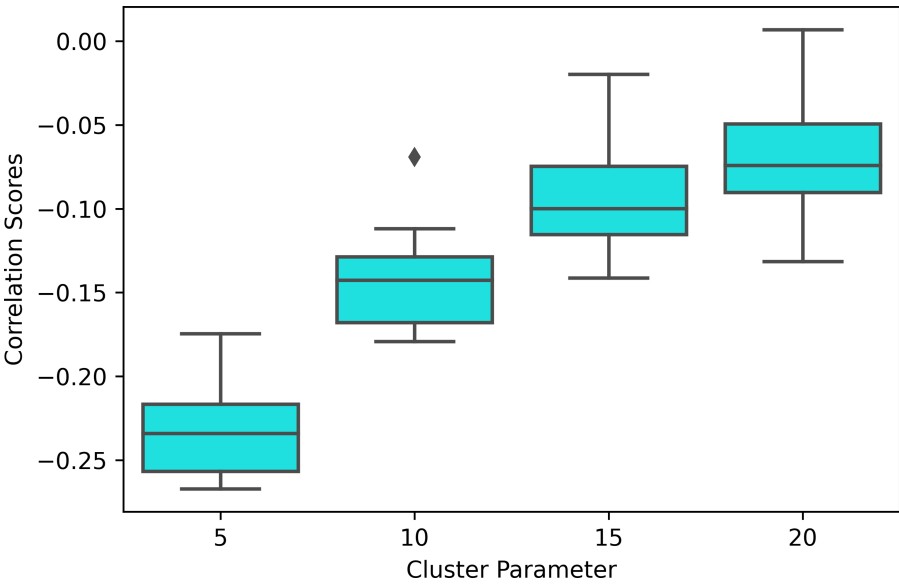

**Figure 13** **Correlation box plot.** Correlation between the modularity score of the Mapper graphs and the response time in seconds across the cluster parameters.

driven by the alpha band, while different dynamics might occur in the other bands. Future work is required to do the same analysis after filtering data in various bands.

The ultimate aim of this research is to extract novel biomarkers to be used in translational studies given high temporal dimensional, minimally processed MEG (or EEG) data sets

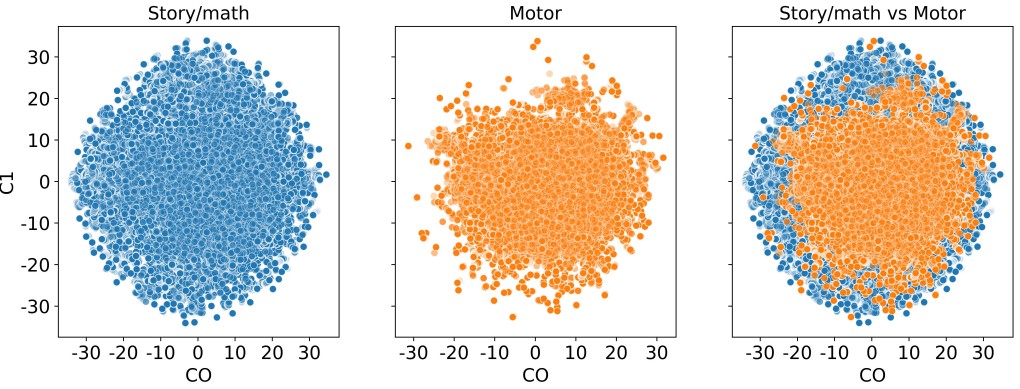

**Figure 14 Dimension reduction by t-SNE.** Scatter plot of the dataset from subject 106521 with story/-math and sensory-motor time points after reducing the dimension to two by t-SNE and before the Mapper algorithm is applied.

of individuals. For example, brain networks extracted from EEG of ADHD group have a significantly lower clustering coefficient and longer characteristic path length than the ones of the control group (*Jang, Kim & Kim, 2020*). Hence, Mapper graphs of the individuals with ADHD traits are expected to have lower centrality measures than normal individuals. On the other hand, it is argued by *Saggar et al. (2018)* that higher centrality results are anticipated from the data of depressed patients compared to healthy individuals, as depressed patients show more functional connectivity than their healthy counterparts using seed-based connectivity approach (*Berman et al., 2011*).

## CONCLUSIONS

Using the graph theoretical invariants of Mapper graphs, we found that the centrality scores of the working memory task are significantly higher than the centrality scores of the story/math task; and the centrality scores of the sensory-motor task are significantly higher than the centrality scores of the story/math task. These results suggest that whole-brain activation patterns are more similar for tasks requiring higher cognitive effort when compared to periods of rest. Likewise, it is demonstrated that there is a weak negative and non-significant correlation between the community structure of the graph and the response time for the working memory and story/math tasks. Although the high number of time points and associated noise contributed to the weak correlation and its non-significance in MEG, this result along with the similar fMRI result has a potential to be improved as the individuals with a specific whole-brain organization are expected to have faster reaction times. The present study shows the potential contribution of the topological data analysis method (*i.e.,* Mapper) to translational studies, while the resulting interactive graphs reveal brain reconfiguration of different frequencies in noisy high-temporal-resolution MEG data at an individual level without losing any information by spatiotemporal collapsing.

As a future direction, it will be interesting to investigate how sensitive the Mapper algorithm is to spatiotemporal collapsing and preprocessing steps of MEG data. Specifically,

it is essential to compare concatenated MEG data with continuous MEG data using the Mapper approach, even though they earlier revealed similar qualitative and quantitative results during rest and task (*Fair et al., 2007*; *Gavrilescu et al., 2008*; *Cheng et al., 2015*; *Zhu et al., 2017*).

Another concern in the current study is applying Mapper on channel-level MEG data recorded by magnetometers; as different source configuration can produce similar MEG channel-level maps, while similar source level data can produce different channel-level signals due different head positions. To address this issue, future studies should investigate application of Mapper on a set of nodes in the source space.

Another direction is to apply Mapper to a set of nodes in the source space rather than to channel-level MEG data recorded by magnetometers. This will address the concern about the fact that different source configuration can produce similar MEG channel-level maps, while different head positions could produce different channel-level patterns from similar source level patterns.

In this study, we have chosen Euclidean distance as a similarity measure and t-SNE as a Mapper filter function. Inspired by manifold learning (*Tenenbaum, Silva & Langford, 2000*), the use of geodesic distances in the mapper algorithm has been recently suggested as the geodesic distances preserve the locality of the original high dimensional data better than Euclidean distance after dimension reduction (*Saggar et al., 2022a*; *Saggar et al., 2022b*). Moreover, the filter function t-SNE can produce artificial clusters in low dimensions. Hence, exploring different similarity measures and filter functions in a future study can provide different insights from the MEG data.

One should also investigate the relationship between mapper graphs and MEG/EEG microstates in cognitive tasks. It is expected that the limited number of microstates are highly connected in mapper graphs due to their similar topography.

## ACKNOWLEDGEMENTS

The authors acknowledge the referees for their insightful comments. The authors would also like to express their appreciation to Linda Geerligs for the valuable discussions during the planning and development of this article. Part of this research was conducted during a visit of AET to Donders Institute, Netherlands.

### Funding

The authors received no funding for this work.

### Competing Interests

The authors declare there are no competing interests.

### Author Contributions

- Ali Nabi Duman conceived and designed the experiments, performed the experiments, analyzed the data, prepared figures and/or tables, authored or reviewed drafts of the article, and approved the final draft.

- Ahmet E. Tatar performed the experiments, analyzed the data, prepared figures and/or tables, authored or reviewed drafts of the article, and approved the final draft.

## Data Availability

The data is available at Zenodo: Ali Nabi Duman. (2023). Processed MEG data used for revealing dynamic brain reconfiguration using MAPPER [Data set]. Zenodo. https://doi.org/10.5281/zenodo.7883031.

The code is available in the Supplemental File.

## Supplemental Information

Supplemental information for this article can be found online at http://dx.doi.org/10.7717/peerj.15721#supplemental-information.

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
