# Peer review of "Topological data analysis for revealing dynamic brain reconfiguration in MEG data"

_PeerJ, doi:10.7717/peerj.15721_

## Round 0.1 · original submission · Major Revisions

Please revise your manuscript as per the comments of both peer reviewers.

Reviewer 1 ·

Basic reporting

The manuscript is generally well-written and shows novelty to some extent. However, the literature review section should be developed. So, the author can add the new articles published between 2020-2022. I suggested that the authors review general and specific applications of neural network models in which they have applied in many fields including: “https://doi.org/10.1063/5.0132846”, and “https://doi.org/10.1038/s41598-022-19157-w”.

Experimental design

This research falls within the scope of the journal. The methods are well-controlled and implemented in the paper.

Validity of the findings

The authors used enough techniques to validate their results.

Additional comments

Comments:
1- The literature review section should be developed. So, the author can add the new articles published between 2020-2022. I suggested that the authors review general and specific applications of neural network models in which they have applied in many fields including: “https://doi.org/10.1063/5.0132846”, and “https://doi.org/10.1038/s41598-022-19157-w”.
2- The authors should expand the conclusion of their work and mention all the essential findings.
3- Can the authors give a discussion about the future work of this study?

Reviewer 2 ·

Basic reporting

The article is well written with acceptable grammar, as well as the revision of the literature. Likewise, the topic is interesting for the neuroimaging community as I see it as an extension of Saggar et al. (2018) research based on fMRI data.

However, in the introduction, some critical issues are not discussed as the negative effect of volume conduction for functional connectivity (FC) analysis, which interpretation can also be affected when FC measures are calculated directly from EEG/MEG channel data. This is less important if this research focuses on discrimination between different mental states or cognitive tasks.

Besides, it is critically observed that in the introduction, the authors pay special attention to discussing FC research, even considering dynamic FC approaches such as those based on sliding widow and hidden Markov model approaches. Notice that If the dimensional reduction conducted in this study with Mapper is performed along the rows (temporal bins), i.e., the dimensional reduction and clustering are applied for the temporal bins, then this approach is more related to the detection of microstates (check these few references: Khanna et al., 2015; Michel et al., 2018; Milz et al., 2016), and thus the authors should consider to revise the latter in the introduction.

References:
[1] Khanna, A., Pascual-Leone, A., Michel, C. M., & Farzan, F. (2015). Microstates in resting-state EEG: current status and future directions. Neuroscience & Biobehavioral Reviews, 49, 105-113.
[2] Michel, C. M., & Koenig, T. (2018). EEG microstates as a tool for studying the temporal dynamics of whole-brain neuronal networks: a review. Neuroimage, 180, 577-593
[3] Milz, P., Faber, P. L., Lehmann, D., Koenig, T., Kochi, K., & Pascual-Marqui, R. D. (2016). The functional significance of EEG microstates—Associations with modalities of thinking. Neuroimage, 125, 643-656

Experimental design

Major issues:

One of the central claims of this and Sagaar et al. (2018) articles is that the Mapper approach avoids collapsing spatial and temporal information. For example, in line 95, the authors state, “Mapper … has the potential to extract clinically relevant information at an individual level without temporal and spatial collapse.”. From my perspective, this is misleading at least …, then what is collapsed when the high dimensionality is reduced first to 2 or 3 dimensions and then represented by a graph? I suggest the authors have a more critical discussion about this.

In line 88, it is claimed that “results partially agree with the fMRI results of Saggar et al. (2018).” but the problem here is that for the referred result, the supporting findings in Sagaar et al. (2018) are also weak. Notice in Sagaar and colleagues’ Fig. 3b (middle) that the negative correlation is weak (r = –0.44, p = 0.071). p=0.071 is not significant. Even more, all those effects shown in that figure are insignificant when considering a control method (e.g., Bonferroni) for multiple comparison tests.

Another possible limitation of techniques such as Mapper is that some information may be lost during the dimensional reduction step, and some structures can be falsely created. For example, using stochastic neighbour embedding (SNE) can find clusters that “even appear in non-clustered data” (https://en.wikipedia.org/wiki/T-distributed_stochastic_neighbor_embedding), which is a serious limitation that should be at least mentioned or discussed in the article.

Validity of the findings

Not clear what information is represented in Figure 6. I imagine the x-axis represents the subjects in the dataset, but what is the information contained in the boxplots? I guess that it may be the modularity score for each Mapper graph’s node. This figure is not well described and discussed in the main text. Therefore, the observations in points 1-3 (lines 344-351) are difficult to understand

Additional comments

Minor issues:
1) No need to make a figure to show the results of the normality test (Figures 4 and 5). Reporting the statistics and p values is sufficient for the report.
2) The authors have a serious issue with the use of parentheses around the cited references.
3) The figure captions are inserted above instead of the traditional location below the images.

---

## Round 0.2 · Minor Revisions

Dear Authors, Please do these revisions:- Some minor details remain, as discussed below by Reviewer 2.

Reviewer 1 ·

Basic reporting

The authors present clear manuscript.

Experimental design

The methodology is described with sufficient details.

Validity of the findings

Conclusions are well-stated and are supported by the results.

Additional comments

I would like to thank the authors for addressing the comments and revision of the manuscript. I suggest the publication of the manuscript.

Reviewer 2 ·

Basic reporting

The authors have addressed most of the concerns in the previous revision. Some minor details remain, as discussed below:

1) Line 38-39: "Recently introduced non-invasive electrophysiological techniques ..."

What is recent here? EEG was introduced in the 1930s by Hans Berger, who demonstrated the measurement of alpha rhythms for the first time, and the MEG technique was also developed long ago in the 1960s.

Please, rewrite this in a more clear sentence.

2) Line 106: "an evidence that A greater."

I imagine "A" in uppercase is a typo here, right?

3) Line 168: "sampling rate is lowered to 506.6275"

Please, mention the unit to make it more clear, e.g., 506.6275 Hz.

4) Lines 235-236: "The input data to Mapper is the (# time points)×(# channels) dimensional matrix prepared by the preprocessing explained in Sec."

This sentence is too short, and also here, as in other parts of the manuscript, when the authors mention "in Sec", it is not clear if "Sec" means "section" to which section you are referring. Please, mention it explicitly.

5) Line 297: "time points of the subject 106521"

Please, use "analysis" instead of "subject" to avoid misinterpreting this with the fact that you may be showing the results for a particular participant data.

I did not realize this notation was clearly explained in the manuscript. It should be over-emphasized as notation for more clarity. I would also suggest a more clear notation, e.g. "i=10,c=30,o=5" or more compactly "i10c30o5" instead of the less informative notation "10305" used in the manuscript to refer to the "interval parameter = 10, cluster parameter = 30, overlap percentage = 5", which can also be confused with other combinations, e.g. 103-0-5.

6) Line 416-417: "it is hard to detect the rapid changes in neural activity using hemodynamic signals from fMRI, as they are the proxy of neural activity"

It should be "a proxy" instead of "the proxy", as fMRI signals are one of the many proxies of neural activity.

7) Figure 7. The quality of this figure should be improved

Instead of plotting "Ordered values" vs "Theoretical quantiles", authors should have inverted the axis, that is, plotting "Theoretical quantiles" vs "Ordered values"

8) Figure 8. The quality of this figure should be improved

The authors must use a horizontal box plot instead of the current version, which is a counter-clockwise rotation of a vertical box plot.

Experimental design

no comment

Validity of the findings

no comment

Additional comments

no comment

---

## Round 0.3 · accepted · Accept

Thank you for your revised manuscript that has been accepted.

Reviewer 2 ·

Basic reporting

I thanks the authors for carefully addressing my concerns. I am satisfied with the current version.

Experimental design

No comment.

Validity of the findings

No comment.

Additional comments

No comment.